# Spontaneous liver disease in wild-type C57BL/6JOlaHsd mice fed semisynthetic diet

**Onne A. H. O. Ronda**[1], **Bert J. M. van de Heijning**[2], **Alain de Bruin**[1,3], **Rachel E. Thomas**[3], **Ingrid Martini**[4], **Martijn Koehorst**[4], **Albert Gerding**[4], **Mirjam H. Koster**[1], **Vincent W. Bloks**[1], **Angelika Jurdzinski**[1], **Niels L. Mulder**[1], **Rick Havinga**[1], **Eline M. van der Beek**[1,2], **Dirk-Jan Reijngoud**[1], **Folkert Kuipers**[1,4], **Henkjan J. Verkade**[1] *

**1** Department of Pediatrics, University of Groningen, University Medical Center Groningen, Groningen, The Netherlands, **2** Danone Nutricia Research, Uppsalalaan, Utrecht, The Netherlands, **3** Dutch Molecular Pathology Center, Faculty of Veterinary Medicine, Utrecht University, Utrecht, The Netherlands, **4** Department of Laboratory Medicine, University of Groningen, University Medical Center Groningen, Groningen, The Netherlands

* h.j.verkade@umcg.nl

**Data Availability Statement:** All relevant data are within the paper and its Supporting Information files.

## Abstract

Mouse models are frequently used to study mechanisms of human diseases. Recently, we observed a spontaneous bimodal variation in liver weight in C57BL/6JOlaHsd mice fed a semisynthetic diet. We now characterized the spontaneous variation in liver weight and its relationship with parameters of hepatic lipid and bile acid (BA) metabolism. In male C57BL/6JOlaHsd mice fed AIN-93G from birth to postnatal day (PN)70, we measured plasma BA, lipids, Very low-density lipoprotein (VLDL)-triglyceride (TG) secretion, and hepatic mRNA expression patterns. Mice were sacrificed at PN21, PN42, PN63 and PN70. Liver weight distribution was bimodal at PN70. Mice could be subdivided into two nonoverlapping groups based on liver weight: 0.6 SD 0.1 g (approximately one-third of mice, small liver; SL), and 1.0 SD 0.1 g (normal liver; NL; p<0.05). Liver histology showed a higher steatosis grade, inflammation score, more mitotic figures and more fibrosis in the SL *versus* the NL group. Plasma BA concentration was 14-fold higher in SL (p<0.001). VLDL-TG secretion rate was lower in SL mice, both absolutely (-66%, p<0.001) and upon correction for liver weight (-44%, p<0.001). Mice that would later have the SL-phenotype showed lower food efficiency ratios during PN21-28, suggesting the cause of the SL phenotype is present at weaning (PN21). Our data show that approximately one-third of C57BL/6JOlaHsd mice fed semisynthetic diet develop spontaneous liver disease with aberrant histology and parameters of hepatic lipid, bile acid and lipoprotein metabolism. Study designs involving this mouse strain on semisynthetic diets need to take the SL phenotype into account. Plasma lipids may serve as markers for the identification of the SL phenotype.

## Introduction

Inbred mouse strains are frequently used in biomedical research, often in combination with standardized (semisynthetic) diets. The rationale behind these choices is to minimize genetic

**Funding:** Danone Nutricia Research provided funding for this study in the form of a grant used to cover bench fees and salary support for OR, as well as in the form of salaries for BvdH and EMvdB. The specific roles of these authors are articulated in the 'author contributions' section. The funder had no role in study design, data collection and analysis, or decision to publish.

**Competing interests:** The authors have read the journal's policy and have the following competing interests: OR received salary and bench fee support via a grant awarded by Danone Nutricia Research. BvdH and EMvdB are employees of Danone Nutricia Research. HJV was a consultant for Danone Nutricia Research outside the submitted work, for which his institution was financially compensated. This does not alter our adherence to PLOS ONE policies on sharing data and materials. There are no patents, products in development or marketed products associated with this research to declare.

**Abbreviations:** PN, postnatal day; FAME, fatty acid methyl ester; MUFA, monounsaturated fatty acid; PUFA, polyunsaturated fatty acid; NEFA, non-esterified fatty acids; AST, aspartate transaminase; ALT, alanine transaminase.

and environmental heterogeneity in experimental models of human diseases. The widely available C57BL/6 mouse strain is the most commonly used inbred mouse strain in basic research [1]. It has offered scientists a powerful tool for understanding gene function and mechanisms of many diseases [2]. Several substrains of C57BL/6 exist, including J, JArc, JRccHsd, JOlaHsd, JBomTac, N, NCrl, NHsd and NTac, which differ genetically and phenotypically [3]. The American Institute of Nutrition (AIN) established guidelines for formulating standardized diets for laboratory mice and rats, to reduce the variation inherent to grain-based and natural-ingredient-based (chow) diets [4]. Diets formulated by the AIN, or derivatives thereof, are widely used in metabolic and nutritional research. Recently, however, by chance we observed a spontaneous bimodal variation in liver weight and very low density lipoprotein (VLDL)-tri-glyceride (TG) secretion rate in a frequently used mouse strain (C57BL/6JOlaHsd) fed a standard semisynthetic control diet (AIN-93G) [5]. We observed that, under pair-housed conditions, each cage tended to contain a mouse with a normal VLDL-TG secretion rate and a mouse with much lower VLDL-TG secretion rate. Mice appeared to have a normal or a small liver size (in wet weight as well as relative to body weight), respectively [5]. We hypothesized that the phenotypic variability in VLDL-TG secretion rate, and possibly also liver weight, was related to social dominance ranks [6]. To investigate the influence of social dominance ranks on VLDL-TG secretion, we subsequently studied this relationship in single- *versus* pair-housed mice [7]. Social housing conditions (single *versus* pair) and thereby social hierarchy appeared, however, not to influence VLDL-TG secretion rate, nor the liver weight [7]. In the present study, we characterized in more detail the spontaneous variation in VLDL-TG secretion rate and liver weight in male C57BL/6JOlaHsd mice fed a standard semisynthetic diet in an attempt to deduce its origin.

## Materials and methods

### Animals

According to national regulations, studies were approved by an external independent national animal experiment committee (CCD, Central Animal Experiments Committee; Centrale Commissie Dierproeven, The Netherlands), after positive advice by the local institutional Committee for Animal Experimentation of the University of Groningen. Subsequently, the study designs were approved by the local institutional Committee for Animal Experimentation of the University of Groningen (Dierexperimentencommissie, protocol number 16–484). Procedures complied with the principles of good laboratory animal care following the European Directive 2010/63/EU for the use of animals for scientific purposes. All efforts were made to minimize animal suffering. Among these efforts, anesthesia (isoflurane & $O_2$) was used during procedures which may cause discomfort or suffering (*e.g.* blood draw, intraperitoneal injection, intragastric administration and termination). All animals were kept in the same temperature-controlled room (20–22˚C, 45–65% humidity, lights on 8AM-8PM) in type 1L (360 cm$^2$) polysulfone cages bearing stainless-steel wire covers (UNO BV, the Netherlands), with wood shaving bedding, Enviro-dri® (TecniLab, The Netherlands) and cardboard rolls. All mice were housed conventionally and handled by the same researcher. All mice were fed diets compliant with AIN-93G (Research Diets Inc. USA, or Ssniff Spezialdiäten GmbH, Soest, Germany) and tap water *ad libitum*.

Breeder C57BL/6JOlaHsd mice were commercially obtained from a specific-pathogen free colony (Envigo, The Netherlands) to be used in this study. Separate batches of breeders were ordered from a commercial supplier over the years 2016–2018. Each batch was used separately to breed pups to be used for the experiments described in this study. The separate 'batches' of breeders were, in our view, a fair representative of what one may expect from a commercial

animal supplier. The various cohorts were subsequently reared, maintained and sacrificed. Virgin C57BL/6JOlaHsd breeders (12 weeks of age at delivery, Envigo, The Netherlands) were acclimatized for at least 2 weeks at our facility and were fed AIN-93G from arrival onward. C57BL/6JOlaHsd breeders were fed AIN-93G during breeding, pregnancy and lactation. Breeders were housed in groups (females), or individual (males) prior to breeding, and were mated in 2-3F+1M groups. The breeding paradigm used in this study was similar to that applied in earlier studies by us [8, 9]. Males were removed after 2 days. Pregnancy was confirmed by a >2 g increase in body weight after 1 week. Upon confirmation of pregnancy, females were housed individually and were not bred again. Only the $F_1$ offspring were used in this study. Nonpregnant females were mated again the following week, for a maximum of 4 times or until pregnancy was confirmed. Each female breeder was used a maximum of one pregnancy. Delivery day was recorded as postnatal day (PN) 0. Pups were randomized between dams, and litters were culled to 4M+2F at PN2, weaned at PN21 and males were housed either in pairs or solitarily (solo). Breeders and female offspring, not further used in this study, were terminated ($CO_2$) at weaning. A C57BL/6J control cohort was bred from the colony maintained at our local animal facility (Central Animal Facility, University of Groningen).

## Study design

The term 'cohorts' refer to a group of mice sacrificed on the same postnatal day. At weaning (PN21), C57BL/6JOlaHsd mice were pair-housed (entire PN42 cohort, entire PN63 cohort, pair-housed PN70 cohort) or solo-housed (solo-housed PN70 cohort only). Mice were sacrificed at PN21, PN42, PN63 or PN70. In a subset of the PN70 cohort, VLDL-TG secretion was determined (see below). S7 Fig shows a visual representation of the study design. Reported 'n = ' represent total number of mice within a cohort, unless specified to be 'SL' or 'NL'. Pair housed C57BL/6J control mice, bred and kept at the Central Animal Facility of the University of Groningen, were fed low-fat semisynthetic control diet (D12450J) during PN56-140. The calculated macro- and micronutrient contents of the diets are given in S4 Table. The fatty acid composition of the AIN-93G diet is provided in S3 Table.

## Body composition

Lean and fat mass were quantified by time-domain nuclear magnetic resonance (LF90II, Bruker Optics, Billerica, MA). Mice were placed into a semi-transparent plastic tube, which was subsequently placed into the bore of the NMR machine. The tube minimized body movements. Body composition analysis took approximately 90 seconds per mouse. Body composition analysis did not require fasting or anesthesia [9]. Simultaneously, body weights and food weights were recorded. Measurements were performed from PN21 once per week until termination at PN70.

## Efficiency of food conversion

Food intake was determined by weekly weighing the food and calculating the difference. The mice were weighed weekly. The <u>e</u>fficiency of <u>c</u>onversion of <u>i</u>ngested food to unit of body substance (ECI) was calculated (opposite to normal, for reasons of visual presentation) as ΔBW (g per mouse per week) / food intake (g per mouse per week).

## VLDL-TG secretion

At PN70, in a subset of the 'PN70 cohort' (n = 14 pair-housed, n = 8 solo-housed), upon intraperitoneal injection of the lipoprotein lipase inhibitor poloxamer-407 (BASF, Ludwigshaven,

Germany) [10] (1 g/kg BW in ~200 μl sterile PBS), retro-orbital blood was drawn at 0, 1, 2, 3 & 5 h in 9 h fasted (midnight-9AM, food taken away at midnight) mice. Blood was drawn during the light-phase. In total, less than 0.2 ml blood was withdrawn per mouse. After the last time point, mice were anaesthetized (isoflurane & $O_2$) and sacrificed by heart puncture and cervical dislocation. A terminal blood sample was collected. Triglycerides (TG) were measured enzymatically (see below). The VLDL-TG secretion rate was calculated from the plasma TG slope over 5 h (mM.h$^{-1}$) multiplied by the estimated plasma volume (28.17 ml/kg BW) [10, 11], giving the TG secretion rate (μmol.h$^{-1}$.kg BW$^{-1}$). It was assumed that all mice had approximately the same plasma volume. Poloxamer 407 was assumed to have a strong effect on gene expression. Thus, tissues obtained from the VLDL-TG secretion cohort were not used for further *postmortem* analyses.

## Termination

At PN21 unfasted mice were anaesthetized at 9AM and sacrificed by heart puncture (n = 12). At PN42, fasted mice (9AM-1PM) were anaesthetized (isoflurane & $O_2$) and sacrificed by heart puncture (n = 14). At PN63 (n = 14) and PN70 (n = 14 pair, n = 8 solo-housed), fasted (midnight-9AM) mice were anaesthetized (isoflurane & $O_2$) and sacrificed by heart puncture. A terminal blood sample was collected. It was assumed that our primary readout parameter (liver weight) would not be largely affected by a duration of fasting up to 9 h. At PN21, pups were housed with their dam and may have been breastfed prior to termination. To minimize animal suffering or discomfort, we chose not to wean the pups prior to PN21 to allow for fasting [12]. At PN42, given the still relatively young age of the mice, we chose a fasting period of 4 h to minimize animal suffering or discomfort. In adult mice (at PN63 and PN70), we chose a standard 9 h fasting period as it was not expected that this would cause unacceptable levels of animal suffering or discomfort. Blood samples were centrifuged at 2,000×*g* for 15 minutes at 4˚C in a tabletop centrifuge (Eppendorf, Nijmegen, the Netherlands). Liver weights were recorded. Whole tissues were snap-frozen in liquid $N_2$ and later cryogenically crushed to powder using a mortar and pestle.

## Assays

Plasma was analyzed using commercially available kits for triglycerides (TG, Roche, Mijdrecht, the Netherlands, cat.no. 11877771216), total cholesterol (TC, Roche, cat. no. 11491458–216), free cholesterol (FC, Spinreact, Santa Coloma, Spain, cat. no. 41035), non-esterified fatty acids (NEFA, Sopachem, 157819910935), phospholipids (Sopachem, Wageningen, the Netherlands, cat. no. 157419910930), aspartate transaminase (AST, Spinreact, cat. no. 1001165), and alanine transaminase (ALT, Spinreact, cat. no. 1001175). Esterified cholesterol was calculated from the difference between total and free. Total plasma protein was determined using a commercial BCA protein assay (Pierce, Thermo Fisher, Landsmeer, the Netherlands).

## Plasma bile acids

Plasma bile acid species were quantified by liquid chromatography-mass spectrometry. To 25 μl of plasma, we added a mixture of internal standards (isotopically labeled bile acids). Samples were centrifuged at 16,000×*g* for 10 min in a tabletop centrifuge (Eppendorf, Nijmegen, the Netherlands) and the supernatant was transferred and evaporated at 40˚C under a stream of $N_2$. Samples were reconstituted in 200 μl methanol:water (1:1), mixed and centrifuged at 1,800×*g* for 3 min. The supernatant was filtered using a 0.2 μm spin-filter at 2,000×*g* for 10 min. Filtrates were transferred to vials and 10 μl was injected into the LC-MS/MS system. The LC-MS/MS system consisted of a Nexera X2 Ultra High Performance Liquid Chromatography

system (SHIMADZU, Kyoto, Japan), coupled to a Sciex Qtrap 4500 MD triple quadrupole mass spectrometer (SCIEX, Framingham, MA, USA). Data were analyzed with Analyst MD 1.6.2 software.

## Fatty-acyl chain profiling

Fatty acid methyl esters (FAMEs) were quantified using gas chromatography (GC) [13]. Cryogenically crushed tissues were homogenized in Potter-Elvehjem tubes (Sigma, St. Louis, MO, USA) in ice-cold phosphate buffered saline (PBS, Gibco, Fisher Scientific, Landsmeer, the Netherlands) solution. A known quantity of homogenized tissue, food, or plasma was transferred to Sofirell tubes, and capped with silicone-PTFE septum screw caps. An internal standard (heptadecanoic acid, C17, Sigma, St. Louis, MO, USA) was added. Lipids were transmethylated at 90°C for 4 h in 6 M HCl:methanol (ratio 1:5), liquid-liquid extracted twice using hexane, transferred to a clean tube, dried at 45°C under a stream of $N_2$, reconstituted in hexane (*n*-hexane PA, Merck) and transferred to GC vials (Aluglas, cat. no. 1013679, APG Europe, Uithoorn, the Netherlands) with inserts (Aluglas, cat. no. 1013586, APG Europe) and caps (VWR, cat. no. 548–0085, Amsterdam, the Netherlands). Samples were analyzed by gas chromatography [13]. The GC system consisted of 6890N network gas chromatograph (Agilent, Middelburg, the Netherlands) and was equipped with a HP-ULTRA 1 dimethylpolysiloxane, nonpolar column (50 m length x 0.2 mm diameter, 0.11 μm film thickness; Agilent, Middelberg, the Netherlands). Samples were injected at 275°C using a 7683 ALS autosampler. The initial column temperature was 160°C and was ramped up at 2°C/min to 240°C, followed by a second-stage ramp-up at 10°C/min to 290°C. The carrier gas was helium at 34 kPa prepressure. The flame ionization detector (FID) was set to 300°C and recorded each sample for 1 hour at 1 Hz. Data were integrated and analyzed using Atlas Chromatography Data System software.

## Liver lipids and total protein content

Total lipids were extracted from liver homogenates using the Bligh & Dyer method [14]. In brief, 50 μl liver homogenate was mixed with 750 μl water, 3 ml chloroform:methanol (1:2), 1.2 ml water and 1 ml chloroform (vortexed at every step) and centrifuged 10 min at 1,000×*g* in a swinging bucket centrifuge (Roto Silenta, Hettich, Geldermalsen, the Netherlands). The bottom fraction was transferred to a clean tube, dried at 50°C under a stream of $N_2$, and reconstituted in 1 ml water with 2% triton X-100 (Fisher Scientific, Landsmeer, the Netherlands). From there, using the abovementioned commercially available kits, liver triglycerides, cholesterol and NEFA were quantified. Hepatic protein content was determined from diluted liver homogenates using a commercial BCA protein assay (Pierce, Thermo Fisher, Landsmeer, the Netherlands).

## Acylcarnitine profiling

Using liquid chromatography-tandem mass spectrometry (LC-MS/MS), acylcarnitine species were quantified [15]. To 10 μl plasma, or 50 μl liver homogenate, a mixture of internal standards (isotopically labeled acylcarnitine species) and acetonitrile was added. Samples were mixed and centrifuged (15,000×*g*, 10 min) to precipitate proteins. Supernatant was transferred to GC vials. Samples were analyzed using LC-MS/MS [15]. The LC-MS/MS system consisted of an API 3000 LC-MS/MS system equipped with a Turbo ion spray source (Applied Biosystems/MDS Sciex, Ontario, Canada). Data were analyzed with Analyst and Chemoview software (Applied Biosystems/MSDSciex, Ontario, Canada).

## RNA isolation and gene expression analysis

Using TRI-Reagent (Sigma, St. Louis, MO, USA), total RNA was extracted from cryogenically crushed whole livers. RNA was quantified by NanoDrop (NanoDrop Technologies, Wilmington, DE, USA). RNA integrity was confirmed by observing the 18S and 28S ribosomal RNA bands on 1% agarose gel (Ultra pure agarose, Thermo Fisher) in Tris-acetate-EDTA buffer (Thermo Fisher, Landsmeer, the Netherlands). The cDNA was synthesized using M-MLV (Invitrogen, Breda, the Netherlands) and random nonamers (Sigma, Darmstadt, Germany). cDNA was quantified by relative standard curve method using quantitative real-time PCR [16]. In brief, cDNA was pooled and serially diluted (10x, 20x, 40x, 80x, 160x). The cDNA samples were diluted to a working concentration (20x) prior to relative quantification. Diluted cDNA samples and the serially diluted relative standard curve were pipetted into MicroAmp plates (Thermo Fisher, Landsmeer, the Netherlands). To each well we added either TaqMan fast advanced master mix (Thermo Fisher) or Sybr green master mix (Thermo Fisher). Primer and TaqMan probe (Eurogentec, Luik, Belgium) sequences are given in S2 Table. Plates were run in a Quantstudio 5 real-time PCR system (Thermo Fisher).

## Histological analysis

Liver was formalin-fixed and paraffin-embedded and sectioned. Paraffin sections were stained with haematoxylin and eosin (H&E) for routine histological analysis, with Picosirius Red for detection of fibrillar collagen, and with Ki-67 for mitotic figures. H&E-stained specimens were scored blindly for steatosis, non-alcoholic steatosis (NAS) [17], ballooning [18] and findings were reviewed by a certified veterinary pathologist (AdB). Ki-67-positive hepatocyte nuclei were counted in 5 separate x40 fields by a single assessor. Fibrosis was assessed using Pico Sirius-red and quantified using ImageJ. Small liver sections were stained with Pico-Sirius red. Portal and parenchymal fibrosis was assessed using digital image analysis software (ImageJ). Briefly, two x20 digital photomicrographs, representing between 50 and 100% of total sample surface area (where possible avoiding large, longitudinal vascular structures), were obtained from comparable regions in every section using polarized light. The Images were analyzed using an in-house developed macro that quantified the area of positively stained collagen fibers (fibrosis) in every image.

## Nicotinamide Nucleotide Transhydrogenase (Nnt[C57BL/6J]) genotyping

DNA was extracted using 75 μl alkaline lysis buffer (25 mM NaOH, 0.2 mM $Na_2EDTA$) and 75 μl neutralization buffer (40 mM Trizma-HCl). Diluted DNA was amplified using HOT FIREPol (Bio-Connect 04-25-02015, Huissen, the Netherlands) and 3 primers. Common: 5'– GTA GGG CCA ACT GTT TCT GCA TGA–3', WT: 5'– GGG CAT AGG AAG CAA ATA CCA AGT TG–3', mutant: 5'– GTG GAA TTC CGC TGA GAG AAC TCT T–3' [19] (Eurogentec, Luik, Belgium). DNA was visualized on 2% agarose gel (Ultra pure agarose, Thermo Fisher) and genotypes were determined based on PCR product length quantified using a DNA ladder (NEB N0556S, New England Biolabs Inc., MA, USA). Samples with a band at 743 bp were considered mutant (*Nnt[C57BL/6J]*), whereas samples with a band at 579 bp were considered wild-type (*Nnt wild-type*) [19].

## Statistical analysis

Statistics were performed using SPSS 23 (SPSS Inc., USA) and R (R Core Team, 2018) [20]. Repeated measures were plotted as median and interquartile range. Single-time data are plotted as Tukey boxplots and scatter plots. No data were excluded. Data were not assumed to be

normally distributed, thus tested non-parametrically. Groups were compared using the exact two-sided Mann-Whitney U test. A p<0.05 was considered to indicate rejection of the null hypothesis. A parameter was considered bimodal when the null hypothesis of Hartigan's dip test was rejected and the scatter plot showed 2 distinct clusters. Principal component analysis (PCA, correlation matrix) was restricted to 2 factors and varimax rotated. PCA variables were exported by method regression and plotted. Classical hierarchical cluster analysis was computed using the unweighted pair group method with arithmetic mean (UPGMA) on the Gower's similarity coefficient for mixed data [21].

## Results

### Liver weight and VLDL secretion of C57BL/6JOlaHsd mice fed semisynthetic diet

We noted a bimodal distribution in absolute and relative liver weight distributions irrespective of whether the mice were pair- or solo-housed at postnatal day (PN) 70 (Hartigan's dip test, p<0.05, Fig 1A). Mice could be subdivided into two, nonoverlapping groups based on liver weight: 0.6 SD 0.1 g (small liver; SL, 2.3 SD 0.1%BW) and 1.0 SD 0.1 g (normal liver; NL, 3.4 SD 0.2%BW). Liver weights at PN70 were statistically significant between SL and NL both in wet weights as well as relative to BW (both p<0.001). Values for bodyweight, as well as lean and fat mass at PN70 were unimodal (Fig 1B). Offspring labeled SL and NL appeared randomly distributed across (surrogate) dams and cages. To determine whether the observations on the bimodality of liver weights were consistent we repeated the experimental procedures in a separately bred and reared cohort (the "PN63" cohort). We noted a bimodal distribution in absolute and relative liver weight distributions at PN63 (p<0.05, Fig 1A). VLDL-TG secretion rate was, irrespective of whether the mice were pair- or solo-housed, lower in SL mice both absolutely (-66%, p<0.001) and after correcting for liver weight (-44%, p<0.001), indicating that the lower VLDL-TG secretion rate possibly is, at least in part, a direct consequence of the differences in liver weight in these mice (Fig 1C). We noted that plasma was visually more yellow in SL *versus* NL mice at PN70 (S8 Fig). Based on liver weights, the SL phenotype occurred at similar rates in pair- and solo-housing (5/14, 36% and 3/8, 38%, respectively, the 'PN70' cohort, Fig 1A). The SL/NL phenotype also occurred in the independent PN63 cohort (8/15, 53%). In subsequent analyses we analyzed parameters from pair-housed mice only, unless specifically mentioned.

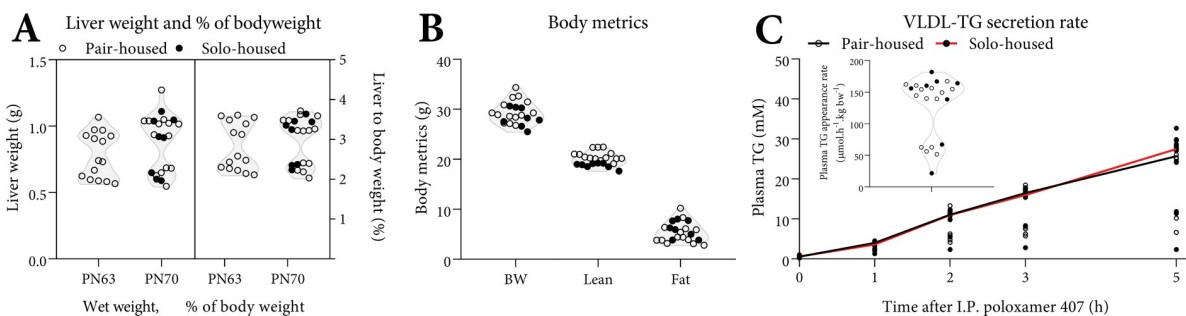

**Fig 1. Liver weight, body weight, body composition and VLDL-TG secretion rate of C57BL/6JOlaHSD mice fed AIN-93G.** Liver weight (A, left), bodyweight, lean tissue and fat tissue (B) at PN70 are expressed in absolute weights and relative liver weight as a percentage of body weight (A, right). Plasma triglyceride (TG) levels and the calculated Very-low density lipoprotein (VLDL)-TG secretion rate ($\mu$mol.h$^{-1}$.kg BW$^{-1}$, inset) expressed as the rate of plasma TG secretion over 5 h after I.P. injection of the lipoprotein lipase inhibitor (Poloxamer 407, 1 g/kg BW, C) at PN70. (C) represents a different cohort than (A-B) (see methods). A: n = 15 (PN63), n = 14 (PN70, pair-housed), n = 8 (PN70, solo-housed). B: n = 14 (PN70, pair-housed), n = 8 (PN70, solo-housed). Data are shown as scatter plots and violin plot. C: n = 14 (pair-housed), n = 8 (solo-housed). Data are shown as aligned scatter plots with the median line. Pair- (○) and solo (●)-housed mice.

## Liver morphology and histological analyses for steatosis, inflammation, proliferation, and fibrosis

Diffuse macroscopic hepatic pallor was exclusively seen in SL mice (Fig 2). Histological analyses revealed notably more severe centrilobular hepatocellular hypertrophy and more frequent karyocytomegaly in SL, as compared with NL mice (p<0.001, Table 1). Ballooning was not seen in either SL or NL livers. SL livers had higher steatosis grades (p<0.05), more necrotic cells and mitotic figures per microscopic field (p<0.001), and more prominent mixed inflammatory cell lobular inflammation (p<0.01) The presence of pigmented macrophages and moderate bile duct hyperplasia was a consistent feature of the SL phenotype (Table 1). Hepatocyte proliferation was higher in SL mice (Fig 2C, Table 1, p<0.01). Moderate liver fibrosis was present in SL mice (Fig 2D, p<0.01). Fibrous depositions were seen around portal triads and central vasculature, and associated with bile duct proliferation. Bridging fibrosis was seen extending from portal to central regions along the sinusoids (particularly evident subcapsularly).

## Hepatic markers of inflammation and fibrogenesis

Expression of inflammatory markers (*Tnf-α*, *Tgf-β*, *Mcp-1*, *F4/80*) was higher in the livers of SL mice. Expression of *Il-1b* and *Il-6*, typically co-elevated with *Tnf-α* (NFκB signaling) was

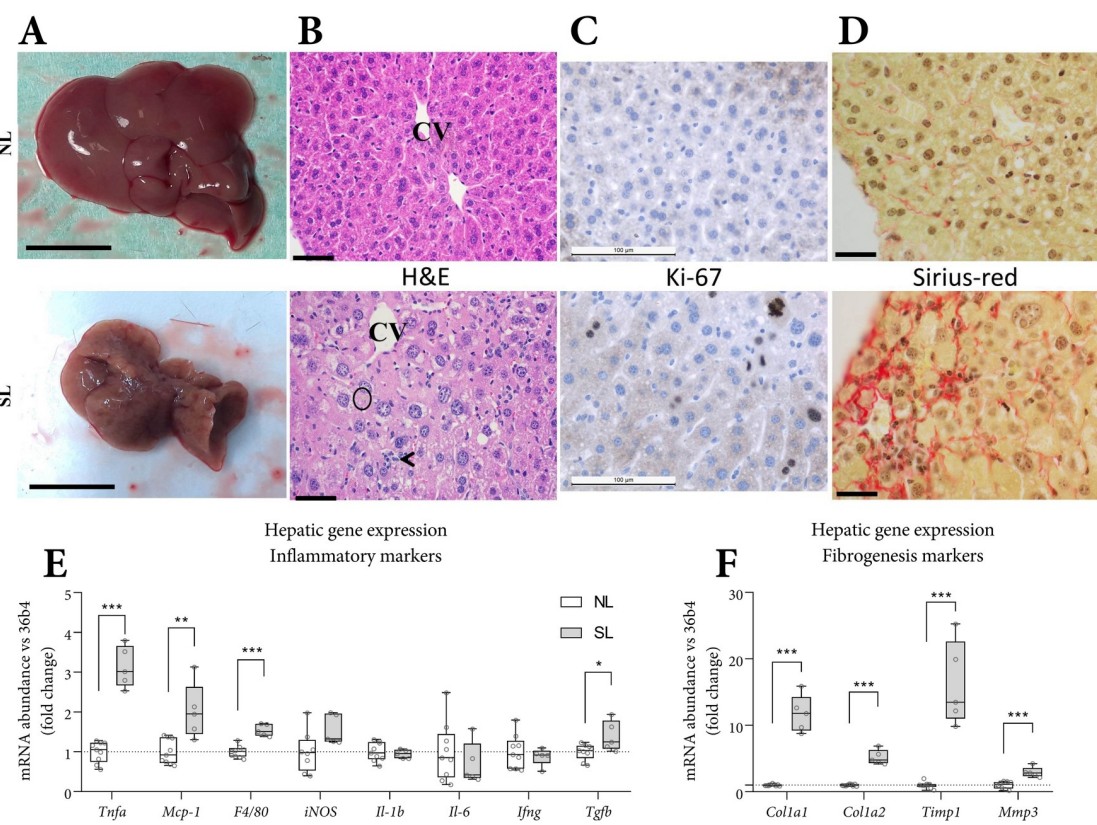

**Fig 2. Liver morphology and histological analyses and gene expression markers of steatosis, inflammation, hepatocyte proliferation, and fibrosis.** Gross liver appearance (most extreme, A, scale bars: 1 cm). Representative histological staining using hematoxylin and eosin (B, 'H&E'), Ki-67 (C, proliferation marker) and Sirius-red (D, collagen fiber staining). Top row: NL, bottom row: SL. Hepatic gene expression for inflammatory (E) and fibrogenesis (F) markers. Histology scale bars: 50 μm. B: CV: central vein, circle: pigmented macrophage, arrowhead: mixed inflammatory cell infiltrate). Histological data represent the PN63 cohort. A-D: NL: n = 7, SL: n = 8. Gene expression data represent the pair-housed PN70 cohort. E-F: NL: n = 9, SL: n = 5. Data are shown as Tukey box plots and scatter plots. Exact two-sided Mann Whitey U test * p<0.05, ** p<0.01, *** p<0.001.

**Table 1. Hepatic histological scoring for steatosis, inflammation, proliferation and fibrosis of NL (n = 7) and SL (n = 8).**

| | | | NL (n = 7) | | SL (n = 8) | | P-value |
|---|---|---|---|---|---|---|---|
| | | | mean | SD | mean | SD | |
| Steatosis | Steatosis grade | *score* | 0.4 | 0.8 | 1.1 | 0.4 | 0.05 |
| Inflammation | Inflammatory foci | *n* | 0.7 | 1.1 | 6.4 | 3.7 | <0.01 |
| | Lobular inflammation | *score* | 0.1 | 0.4 | 1.1 | 0.6 | <0.01 |
| | NAFLD activity | *score* | 0.6 | 0.8 | 2.3 | 0.9 | <0.01 |
| | Pigmented macrophages | *presence* | 0/7 | | 8/8 | | |
| | Sirius red area | *fold change* | 1.0 | 0.4 | 8.0 | 4.5 | <0.01 |
| Proliferation | Karyocytomegaly | *score* | 0.9 | 0.2 | 2.9 | 0.4 | <0.001 |
| | Mitotic figures | *n* | 0 | 0 | 4.6 | 1.8 | <0.001 |
| | Bile duct hyperplasia | *score* | 0.1 | 0.2 | 1.6 | 0.4 | <0.001 |
| | Ki-67 | *n* | 0.3 | 0.2 | 7.3 | 3.7 | <0.01 |

Quantification of Fig 2. Steatosis grade 0 = < 5%; 1 = 5–33%; 2 = 33–66%; 3 = > 66%. Inflammatory foci per 5 (200x) fields. Lobular inflammation 0 = none; 1 = <2 foci; 2 = 2–4 foci; 3 = >4 average foci/200x field. Non-alcoholic fatty liver disease (NAFLD) activity score = sum of steatosis + lobular inflammation + ballooning); score > = 5 steatohepatitis; Score <3 non-steatohepatitis. Ballooning was not seen in any sample. Mitotic figures per 5 (40x) fields. Biliary hyperplasia (0 = none; 1 = mild; 2 = moderate; 3 = marked). Ki-67: average positive hepatocyte nuclei per 5 (200x) fields. Sirius-red area: fold change of Sirius-red-positive signal in comparable surface areas. Data represent the PN63 cohort. Values represent means ± SD. Exact two-sided Mann-Whitney U test.

similar between groups (Fig 2E). Expression of fibrogenesis markers (fibrillar forming collagens *Col1a1* and *Col1a2*, and the (inhibitor of) matrix metalloproteinase *Timp1* and *Mmp3*) were profoundly higher in SL mice (Fig 2F).

## Hepatic triglyceride content, plasma liver enzymes and fasting plasma lipids in SL and NL mice

In accordance with the histological steatosis, we found a higher hepatic triglyceride content in SL livers compared with NL (+105%, Fig 3A). Hepatic free fatty acid content (29 SD 11 versus 16 SD 5 µmol/g, p<0.05) was higher in SL *versus* NL. Plasma liver enzymes ASAT and ALAT (aspartate/alanine aminotransferase) were higher in SL (Fig 3B). Fasting plasma TG and cholesterol levels were lower. Plasma NEFA was subtly higher in SL (Fig 3C), whereas total plasma protein (52 SD 4 *versus* 50 SD 4 mg/ml) concentrations were similar. Hepatic expression of genes related to lipogenesis (*Fasn*, *Srebp1f*, and *Dgat1/2*) were similar. Expression of *Pparg1*, the master regulator of the adipogenic program to store fats in lipid droplets, was higher in SL (Fig 3D).

## Essential fatty acids and long-chain polyunsaturated fatty acids in SL and NL mice

Essential fatty acid deficiency (EFAD) may cause or contribute to liver disease [22]. We assessed EFA status/concentrations by determining the hepatic fatty acyl-chain (FA) profile, primarily defined by moieties of plasma membranes, cholesteryl esters, free fatty acids and triglycerides. The hepatic FA profile at PN70 showed large differences between SL and NL. All ω6 species, apart from 18:2ω6, were higher (p<0.05) in the livers of SL mice (Fig 3E). The hepatic osbond:DHA ratio, a marker of poor DHA status, was higher in SL (Fig 3F). These observations were also found in the separate groups of mice at PN63 (S1H & S1I Fig, respectively). With respect to ω3/6 moieties, we detected 18:3ω3, 18:2ω6 and trace amounts 20:2ω6 in the diet, (S3 Table). The ω3/6 levels were typical for a soybean oil-based diet with no other FA-bearing ingredients, such as the semisynthetic AIN-93G diet. Osbond acid (22:5ω6)

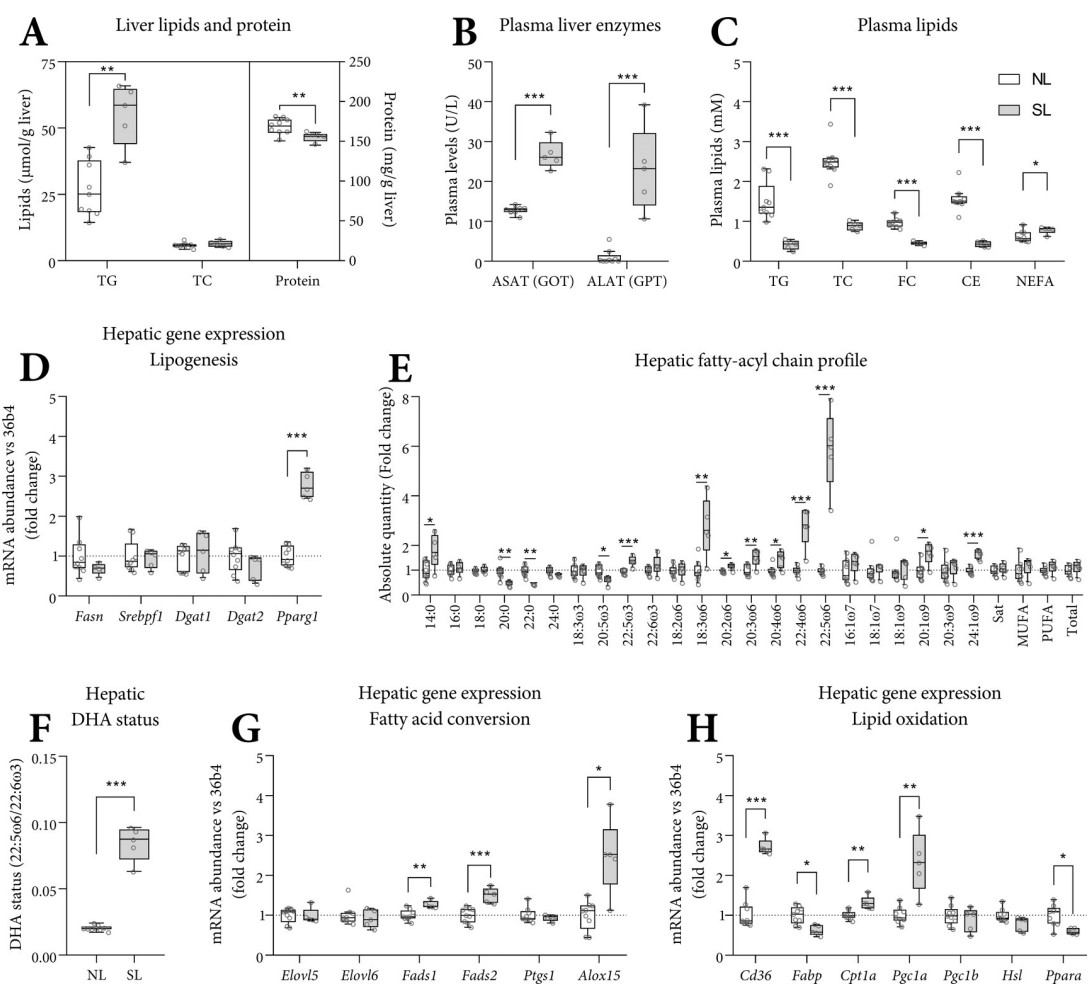

**Fig 3. Plasma lipids, liver enzymes, hepatic lipid metabolism and levels of essential fatty acids and long-chain polyunsaturated fatty acids.** Hepatic TG and total cholesterol (μmol/g) and protein (mg/g) are expressed as absolute concentrations (A). Fasting plasma ASAT (GOT) and ALAT (GPT) are expressed as units/L (B). Fasting triglycerides (TG), total (TC), free (FC), esterified (CE) cholesterol, and non-esterified fatty acids (NEFA, E) are expressed as absolute concentrations. Hepatic gene expression was corrected for *36b4* and shown as fold-change *versus* NL (D, G-H). The hepatic FA profile is shown as the fold-change of absolute values *versus* NL (E). The hepatic DHA status was calculated as the molar ratio 22:5ω6/22:6ω3 (F). Abbreviations used: ASAT/ALAT: aspartate/alanine aminotransferase. 14:0 myristic acid, 16:0 palmitic acid, 18:0 stearic acid, 20:0 arachidic acid, 22:0 behenic acid, 24:0 lignoceric acid, 18:3ω3 α-linolenic acid (ALA), 20:5ω3 eicosapentaenoic acid (EPA), 22:5ω3 docosapentaenoic acid (DPA), 22:6ω3 docosahexaenoic acid (DHA), 18:2ω6 linoleic acid (LA), 18:3ω6 γ-linolenic acid (GLA), 20:2ω6 eicosadienoic acid, 20:3ω6 dihomo-γ-linolenic acid (DGLA), 20:4ω6 arachidonic acid (ARA), 22:4ω6 adrenic acid, 22:5ω6 osbond acid (OsbA), 16:1ω7 palmitoleic acid, 18:1ω7 vaccenic acid, 18:1ω9 oleic acid, 20:1ω9 gondoic acid, 20:3ω9 mead acid, 24:1ω9 nervonic acid. Sat: Σsaturated FAs, MUFA: Σmono-unsaturated FAs, PUFA: Σpoly-unsaturated FAs. A-H: n = 9 (NL) and n = 5 (SL), Tukey box plots and scatter plots. Data represent the pair-housed PN70 cohort. Exact two-sided Mann-Whitey U test *: p<0.05, ** p<0.01, *** p<0.001.

concentration was higher in plasma, heart and skeletal muscle, resulting in a higher osbond: DHA ratio (S1A–S1F Fig). Analyzing the hepatic fatty acyl-chain profiles using principal component analysis (PCA), NL and SL mice from PN63 and PN70 appeared to form discrete clusters based on their NL/SL status, whereas their age (PN63 *versus* PN70) seemingly had no effect on these principal components (S1G Fig). To approximate when the SL phenotype develops, we repeated the experimental procedures in a separately bred and reared cohorts (the "PN21" and "PN42" cohort). Mice from the PN42 cohort also appeared to form two discrete clusters, near the SL and NL cluster, whereas the PN21 cohort clustered separately (S1G

Fig). The molar plasma triene/tetraene ratio, a surrogate EFAD marker, was similar in SL and NL mice at PN70 and 63 (S1B & S1J Fig, respectively). Essential (dietary) fatty acids can be elongated, desaturated and used to synthesize (among others) ligands involved in inflammatory processes. Hepatic expression of elongases *Elovl5*/6 and of cyclooxygenase 1 (*Ptgs1*), were similar between SL and NL mice. Expression of desaturase enzymes *Fads1/2* (Δ5/6 desaturase, D5/6D) and of lipoxygenase 15 (*Alox15*) was higher in SL mice (Fig 3G). Lipids can enter the liver as free fatty acids via transporters including *Cd36*, which was higher expressed in SL (Fig 3H). The expression of markers of β-oxidation was ambiguous; expression of the transporter of long-chain FAs (*Fabp1*) and the master regulator of lipid metabolism (*Ppara*) were lower in SL, whereas that of a member of the carnitine shuttle (*Cpt1a*) and the master regulator of mitochondrial/peroxisomal biogenesis (*Pgc1a*, Fig 3H) was higher in SL.

## Plasma bile acid species in SL and NL mice

Liver disease can coincide with altered bile acid (BA) metabolism [23–25]. In SL mice, at PN70, fasting plasma BA were 14-fold higher than in NL mice (Fig 4A). BA are synthesized by the liver as primary species (cholate: CA, ursodeoxycholate: UDCA, chenodeoxycholate: CDCA and α/β-muricholate: α/β-MCA), and taurine (T) conjugated. In the gut lumen, BA are deconjugated and converted to one of the secondary species (hyo/deoxycholate: H/DCA, lithocholate: LCA, ω-muricholate: ω-MCA) by the microbiota [26]. The sums of the primary BA species (168 SD 49 *versus* 3 SD 4 μM, p<0.001) and the secondary BA species (16 SD 9 *versus* 2 SD 3 μM, p<0.001) were higher in SL. The secondary-to-primary species ratio was lower in SL (0.1 SD 0.1 versus 0.9 SD 0.4, p<0.01) (Fig 4B). The ratio between conjugated and deconjugated BA was similar between groups.

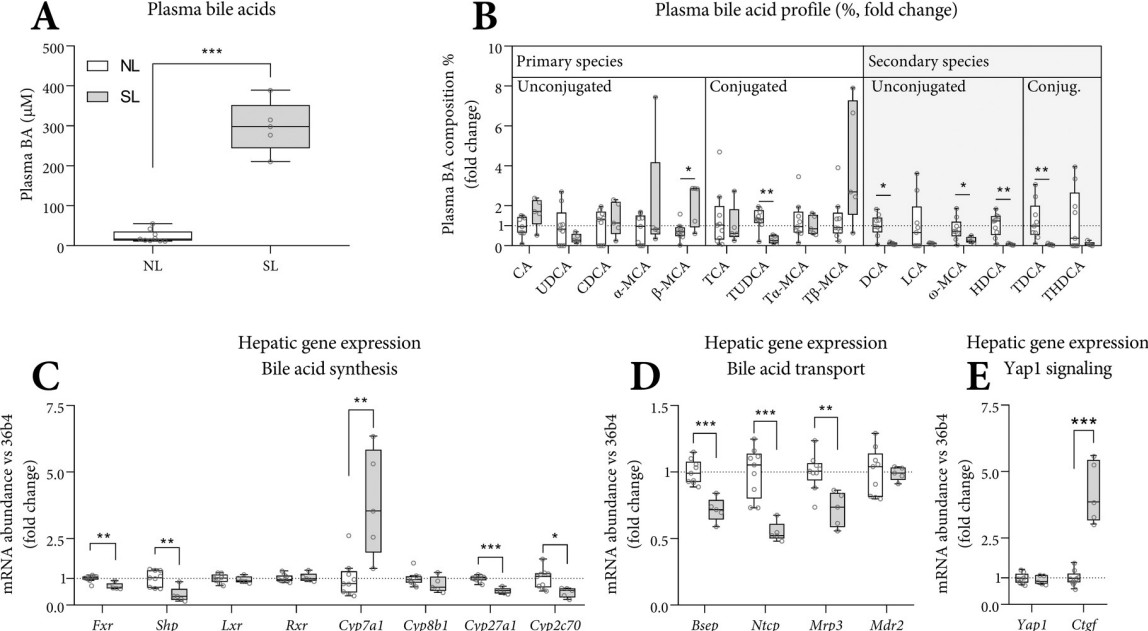

**Fig 4. Plasma bile acids and hepatic mRNA expression of BA synthesis and transport genes in SL and NL mice.** Fasting plasma bile acids (A) are expressed as absolute concentrations. For visual clarity, the percentages of each bile acid species are shown as a fold change *versus* NL (B). Hepatic gene expression was corrected for *36b4* and shown as fold-change *versus* NL (C-E). Abbreviations used: (T) (L) CA: (tauro) (litho) cholate, (T) (U/C/H) DCA: (tauro) (urso/cheno/hyo) deoxycholate, (T) (α/β/ω)-MCA: (tauro-) α/β/ω-muricholate. A-E: n = 9 (NL) and n = 5 (SL), Tukey box plots and scatter plots. Data represent the pair-housed PN70 cohort. Exact two-sided Mann Whitey U test * p<0.05, ** p<0.01, *** p<0.001.

## Hepatic mRNA expression of BA synthesis and transport genes in SL and NL mice

Hepatic *Cyp7a1* expression, encoding the rate limiting enzyme in BA synthesis, was higher in SL mice at PN70. Other bile acid synthesis enzymes *Cyp27a1* and *Cyp2c70* were lower in SL, whereas *Cyp8b1* was similar. The bile acid receptor *Fxr* and its downstream target *Shp* were lower in SL, with similar levels of *Lxr* and *Rxr* (Fig 4C). Expression of hepatic BA transporters (*Bsep*, *Ntcp* and *Mrp3*) was lower in SL, suggesting lower transhepatic BA fluxes. Expression of *Mdr2*, the biliary phospholipid transporter, was similar between groups (Fig 4D). Bile acid metabolism is closely linked to liver size control via Fgf15 and Hippo signaling [23]. Expression of *Ctgf* (YAP target gene [23]) was higher in SL (Fig 4E).

## Hepatic expression markers of cellular and mitochondrial stress

We determined gene expression markers of cellular and mitochondrial stress in hepatic tissue of SL and NL mice at PN70. Hepatic expression of *Ddit3* (often called *Chop*), a cellular stress marker, was higher in SL. Markers of endoplasmic reticulum (ER) stress (*Atf4*, *Atf6*, *Gadd34*, *Trib3*) were similar (Fig 5A). In classic ER-stress, molecular chaperones are upregulated, which was not the case in SL mice, where those were downregulated (Fig 5B). Instead, *Atf5*, the mitochondrial unfolded protein response (UPR) mediator, and *Hmox1*, involved in protection against oxidative stress, were higher in the liver of SL mice.

## Hepatic and plasma acylcarnitine species in SL and NL mice

Higher *Atf5* and *Hmox1* expression (Fig 5A) may indicate mitochondrial dysfunction. We assessed hepatic and plasma acylcarnitines as a surrogate marker for mitochondrial (β-oxidation) function. Hepatic acetylcarnitine (C2) concentration was 27-fold higher in SL *versus* NL. In addition, glutarylcarnitine derivatized conjugate (C5DC) and methylglutarylcarnitine (C6DC, both p<0.001) were 3-fold higher in SL. In plasma, long-chain acylcarnitine species were higher (+74%, C14-18, p<0.001, Table 2) in SL mice. As C57BL/6J mice have a defective nicotinamide nucleotide transhydrogenase (*Nnt*) gene, involved in the generation of the antioxidants glutathione and thioredoxin [19], we genotyped $Nnt^{C57BL/6J}$ in our C57BL/6JOlaHsd mice. It appeared that C57BL/6JOlaHsd mice were wild-type for *Nnt* (S4 Fig).

## Course of SL

For investigating the timeframe wherein the SL phenotype develops, we assessed body weights, body composition and food intake from weaning until PN70. For this, we used the solo-

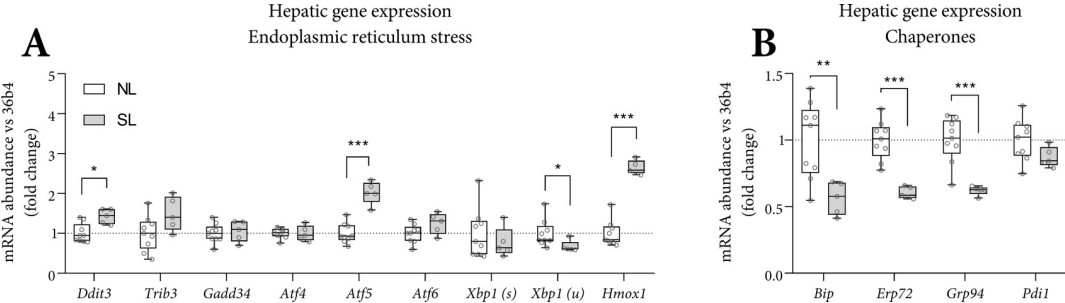

**Fig 5. Hepatic mRNA expression of genes involved in endoplasmic reticulum stress.** Hepatic gene expression for endoplasmic reticulum stress (A) and chaperones (B) was corrected for *36b4* and shown as fold-change *versus* NL. A-B: n = 9 (NL) and n = 5 (SL), Tukey box plots and scatter plots. Data represent the pair-housed PN70 cohort. Exact two-sided Mann Whitey U test *: p<0.05, ** p<0.01, *** p<0.001.

**Table 2. Liver and plasma acylcarnitine species in SL and NL mice at PN70.**

| | | Liver (nmol/g) | | | | Plasma (μM) | | | | |
| --- | --- | --- | --- | --- | --- | --- | --- | --- | --- | --- |
| | | NL (n = 9) | | SL (n = 5) | | P-value | NL (n = 9) | | SL (n = 5) | | P-value |
| | | mean | SD | mean | SD | | mean | SD | mean | SD | |
| Sum | | 265 | 34 | 441 | 34 | <0.001 | 44 | 6 | 38 | 5 | <0.05 |
| Sum C14-C18 | | 0.65 | 0.35 | 0.91 | 0.24 | <0.05 | 1.1 | 0.2 | 2.0 | 0.2 | <0.001 |
| Free/bound ratio | | 9.2 | 1.1 | 5.3 | 1.8 | <0.001 | 10.1 | 1.5 | 6.3 | 0.6 | <0.001 |
| Common name | Abbreviation | | | | | | | | | | |
| L-carnitine | C0 | 233 | 30 | 363 | 16 | <0.001 | 14 | 3.0 | 11 | 2.4 | 0.06 |
| Acetylcarnitine | C2 | 0.4 | 0.2 | 11 | 15 | <0.01 | 27 | 6 | 24 | 3 | 0.06 |
| Propionylcarnitine | C3 | 0.09 | 0.05 | 0.72 | 0.33 | <0.001 | 0.32 | 0.11 | 0.30 | 0.07 | 0.8 |
| Butyrylcarnitine | C4 | 0.07 | 0.02 | 0.45 | 0.16 | <0.001 | 0.21 | 0.05 | 0.18 | 0.02 | 0.2 |
| Tiglylcarnitine | C5:1 | 0.09 | 0.15 | 0.09 | 0.04 | 0.2 | 0.01 | 0.00 | 0.02 | 0.00 | 0.09 |
| Isovaleryl carnitine | C5 | 0.07 | 0.02 | 0.15 | 0.03 | <0.01 | 0.06 | 0.01 | 0.08 | 0.01 | <0.05 |
| Hexanoylcarnitine | C6 | 0.00 | 0.00 | 0.03 | 0.04 | 0.1 | 0.03 | 0.01 | 0.06 | 0.01 | <0.01 |
| Octanoylcarnitine | C8 | 0.11 | 0.05 | 0.13 | 0.05 | 0.5 | 0.03 | 0.01 | 0.03 | 0.01 | 0.7 |
| Decenoylcarnitine | C10:1 | 0.06 | 0.02 | 0.08 | 0.03 | 0.5 | 0.01 | 0.00 | 0.02 | 0.00 | <0.01 |
| Decanoylcarnitine | C10 | 0.07 | 0.00 | 0.19 | 0.03 | <0.001 | 0.01 | 0.01 | 0.02 | 0.00 | 0.09 |
| Dodecenoylcarnitine | C12:1 | 0.19 | 0.07 | 0.11 | 0.08 | 0.06 | 0.02 | 0.01 | 0.03 | 0.01 | <0.05 |
| Dodecanoylcarnitine | C12 | 0.00 | 0.00 | 0.01 | 0.03 | 0.4 | 0.03 | 0.01 | 0.05 | 0.01 | <0.001 |
| Tetradecenoylcarnitine | C14:1 | 0.07 | 0.00 | 0.07 | 0.00 | 1 | 0.09 | 0.02 | 0.17 | 0.04 | <0.001 |
| Tetradecanoylcarnitine | C14 | 0.04 | 0.03 | 0.05 | 0.03 | 1 | 0.11 | 0.02 | 0.27 | 0.05 | <0.001 |
| Hexadecenoylcarnitine | C16:1 | 0.04 | 0.04 | 0.07 | 0.00 | 0.2 | 0.10 | 0.03 | 0.22 | 0.04 | <0.001 |
| Hexadecanoylcarnitine | C16 | 0.23 | 0.31 | 0.25 | 0.07 | <0.01 | 0.34 | 0.06 | 0.51 | 0.03 | <0.001 |
| Octadecadienoylcarnitine | C18:2 | 0.08 | 0.03 | 0.11 | 0.04 | 0.3 | 0.12 | 0.02 | 0.24 | 0.04 | <0.001 |
| Octadecenoylcarnitine | C18:1 | 0.12 | 0.04 | 0.21 | 0.11 | 0.06 | 0.32 | 0.04 | 0.51 | 0.06 | <0.001 |
| Octadecanoylcarnitine | C18 | 0.07 | 0.02 | 0.15 | 0.03 | <0.01 | 0.07 | 0.01 | 0.08 | 0.01 | 0.1 |
| Butyrylcarnitine + Malonylcarnitine | C4OH+C3DC | 4.6 | 0.8 | 5.8 | 1.7 | 0.3 | 0.4 | 0.1 | 0.3 | 0.1 | 0.4 |
| 3-OH-isovalerylcarnitine + Methylmalonylcarnitine | C5OH+C4DC | 0.76 | 0.11 | 1.29 | 0.17 | <0.001 | 0.08 | 0.02 | 0.09 | 0.03 | 0.6 |
| Glutarylcarnitine | C5DC | 17 | 4 | 49 | 14 | <0.001 | 0.34 | 0.07 | 0.51 | 0.21 | <0.01 |
| 3-Methylglutarylcarnitine | C6DC | 2.0 | 0.6 | 6.7 | 0.7 | <0.001 | 0.0 | 0.0 | 0.1 | 0.0 | 0.2 |

Liver (nmol/g) and plasma (μM) acylcarnitine species are shown as absolute values. Values represent means ± SD. Data represent the pair-housed PN70 cohort. Exact two-sided Mann Whitney U test. n.s.: not significantly different between NL and SL.

housed PN70 cohort. Bodyweight (Fig 6A) and lean and fat mass (Fig 6B) suggest that SL mice grow slower immediately after weaning. Food intake was similar, but the food efficiency ratio (BW gain per gram food per week) was lower in SL mice in the first week following weaning (PN21-28, Fig 6C). Relative liver weight was unimodal at PN21 and 42 (Fig 6D). Plasma bile acid levels varied widely in the PN21 cohort (2.7–74 μM), in the PN42 cohort (0.3–176 μM), and in the PN63 cohort (0.2–191 μM; Fig 6E). Analysis of relative liver weight, plasma lipids, bile acids and hepatic fatty acyl-chain profiles using principal component analysis (PCA), showed that NL and SL mice from PN63 and PN70 appeared to form discrete clusters based on their NL/SL status. The age (PN63 *versus* PN70) had no apparent effect on these principal components. The PN42 cohort, which does not display bimodality with respect to wet or relative liver weight, does display 2 clusters on the PCA scatter plot, closely positioned to the NL and SL clusters. Mice from PN21 cluster together on a scatter plot, separate from NL and SL clusters (Fig 6F). Hierarchical cluster analysis of relative liver weights, plasma parameters and hepatic fatty acyl-chain profiles showed that the PN42 cohort contained mice (4/14; 29%) that

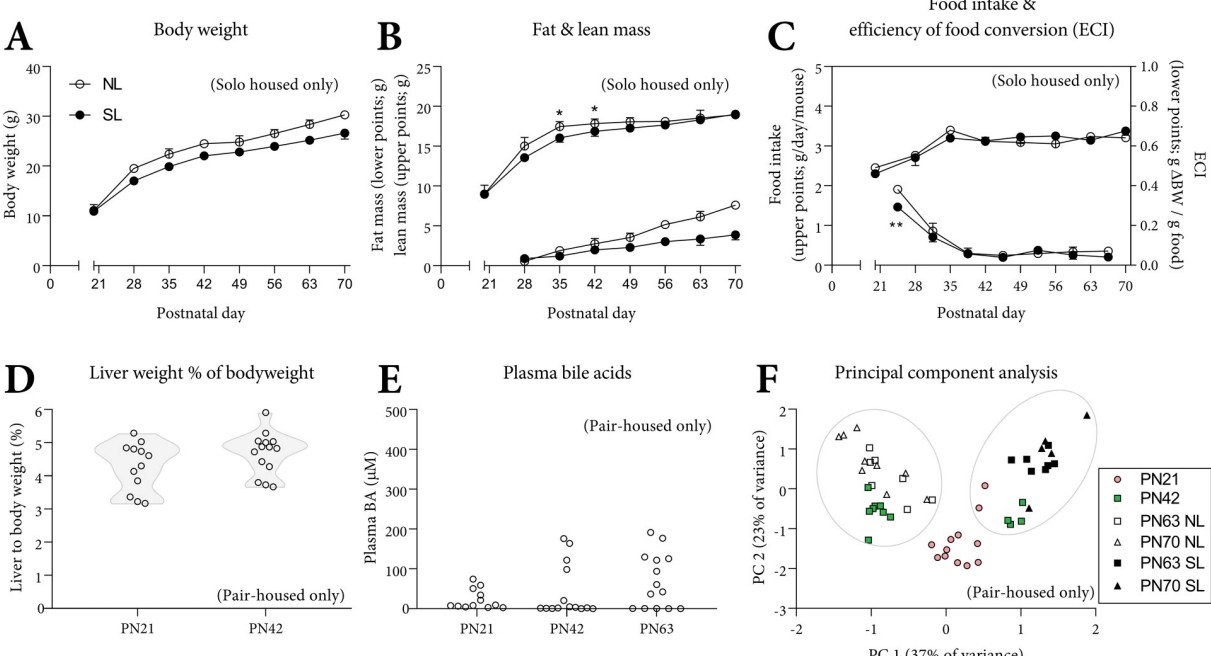

**Fig 6. Body metrics, food intake, liver weight and plasma BA over-time.** Bodyweight (A), lean (B, upper points) and fat mass (B, lower points) are shown as absolute values. Food intake (C, upper points) is shown as absolute values and represents food intake 24 h subsequent to the indicated time (PN21) or during 7 d prior to the indicated time (PN28-70). The efficiency of conversion of ingested food to unit of body substance (ECI, ΔBW/food intake per week) is shown per week (C, lower points). Liver weight as % of bodyweight at PN21, PN42, PN63 and PN70 (D). Absolute plasma bile acid levels at PN21, PN42 and PN63 (E). Principal component analysis (PCA) on relative liver weight, plasma lipids and liver fatty acyl-chain profiles from pair-housed PN21, PN42, PN63 and PN70 cohorts. Principal component (PC) 1 (37% of variance) and PC 2 (23% of variance) on the x- and y-axis respectively (F). A-C: data represent the single-housed PN70 cohort, median and interquartile ranges, error bars are not shown when they fall within the symbol, NL: n = 11, SL: n = 5. D: data represent the pair-housed PN21 (n = 12) and PN42 (n = 14) cohorts. E: Data represent the pair-housed PN21 (n = 12), PN42 (n = 14) and PN63 (n = 14) cohorts. F: Data represent the pair-housed PN21 (n = 12), PN42 (n = 12), PN63 (n = 7 NL, n = 7 SL) and pair-housed PN70 (n = 9 NL, n = 5 SL) cohorts. Exact two-sided Mann-Whitney U test. * p<0.05, ** p<0.01.

clustered with SL mice (cluster 2) whereas others clustered with NL mice (cluster 1), and the PN21 cohort clustered separately (S2 Fig). The two PN42 clusters were analyzed as separate groups. Hepatic mRNA expression of *Col1a1*, *Cd36*, *Pparg1* and *Hmox1* was higher in cluster 2 (S3A and S3B Fig). The hepatic fatty-acyl chain profiles at PN42 (S3C Fig) showed a distinct pattern between cluster 1 and 2. These data suggest that cluster 2 may be (an early form of) the SL phenotype. The hepatic FA profile at PN21 (S3D Fig) was similar for all mice and no distinct clusters could be identified. Plasma triglycerides and cholesterol at PN42 and PN63 were, similarly to PN70, lower in SL *versus* NL mice. Liver acylcarnitines (+47%, p<0.01), in particular C5DC and C6DC (both +200%. p<0.01) were higher in cluster 2 *versus* cluster 1 mice at PN42 (S1 Table). At PN21, all mice had similar plasma lipid levels (S5 Fig). At PN21, all mice had similar liver acylcarnitine profiles (S5 Table). The cohorts (PN42: 29%, PN63: 53%, PN70: 35%) suggest that the SL phenotype occurs in approximately one-third of C57BL/6JOlaHsd mice fed semisynthetic diet. In a control experiment, WT C57BL/6J mice, reared on chow, were fed a low-fat semisynthetic control diet (D12450J, Research Diets Inc. USA) from PN56 until PN140 (S6 Fig). Hierarchical cluster analysis of relative liver weights, plasma lipids and total plasma bile acids indicated that among the 9 mice tested, 2 mice appeared to cluster separately from the remaining 7 mice (S6A Fig). Histological analysis revealed that these 2 mice, compared to the 7 others, had more prevalent karyocytomegaly, mitosis, single cell death and scattered moderate mononuclear infiltration (S6B–S6E Fig).

## Discussion

In this study, we show spontaneous divergence in liver weight in experimentally and nutritionally identically treated C57BL/6JOlaHsd mice fed a commonly used semisynthetic low-fat diet (AIN-93G). This diet is typically not considered to induce a model of liver disease. The divergence in liver weights resulted in mice with a small liver (SL) or normal liver NL) and coincided with profound metabolic differences in terms of lipid and bile acid metabolism. Our data demonstrate that the SL phenotype resembles, to some extent, the biochemical and histological changes observed in human with chronic liver disease, what ultimately may lead to chronic liver failure. Our data indicate that the first aspects of the phenotype become notable at or immediately after weaning.

Heterogeneity and heterogeneous responses to stimuli in inbred mice have been reported in the context of high-fat diet feeding [27, 28], in neurobiology [29, 30], and in the apoE*3-Leiden.CETP model [24, 31]. Clear parallels exist between our study and those performed in apoE*3-L.CETP mice [24, 31], which are empirically subdivided (at PN42) into phenotypical responders (R) and non-responders (NR) based on plasma lipids [24]. Upon reassessing this literature, however, it became apparent that, despite the suggestive name, R and NR mice already differ in plasma lipids on chow [24]. This illustrates that "non-responding" to a dietary challenge is preceded by an existing spontaneous phenotype [24]. NR apoE*3-L.CETP show, similar to our SL, lower liver weight, higher liver TG, an inflammatory liver pathology, higher plasma liver enzymes and higher plasma BA, compared to R or NL mice [24]. It may thus well be that the spontaneous bimodal liver size distribution observed in our study and the (non-)responders in apoE*3L.CETP mice have a similar etiology and, thus, that the phenotype is not facility-dependent nor strain-dependent.

Histologically and based on gene expression patterns, the livers from SL mice showed an inflammatory phenotype and mild steatosis. Marked karyocytomegaly in the livers of SL mice likely indicates polyploidy and higher rate of cell division. The karyocytomegaly was linked to hepatocellular swelling that may reflect glycogen storage or acute hydropic degeneration [32]. Despite the suggested higher hepatocyte division rate, livers of SL are smaller. Thus, higher rates of hepatocyte cell division may be counteracted by, or even compensatory for, higher hepatocyte cell death (degeneration). The presence of pigmented macrophages in the livers of SL mice, together with elevated hepatic *F4/80*, *Tnf-α*, *Mcp-1* and *Col1a1* expression, are suggestive of an immune response which activates fibrogenesis [33]. Mild steatosis was observed in the livers of SL mice, together with lower VLDL-TG secretion rate and higher plasma NEFA and hepatic gene expression of *Cd36*. Hepatic expression and protein levels of Cd36 are higher in, and thought to contribute to, non-alcoholic steatosis [34]. The higher hepatic TG levels, lower VLDL-TG secretion rates, and higher *Cd36* expression levels in SL mice are suggestive of a (net) flux towards hepatic TG stores.

Supplementing DHA in an acute model of liver damage (CCL$_4$) lowers the fibrotic and inflammatory response [22]. The enzymes necessary for the conversion of 18:3ω3 to DHA (ω3) also convert ω6 species and thereby synthesize osbond acid. Higher osbond acid concentrations, and high osbond:DHA ratios have been observed in human NAFLD liver biopsies [35, 36]. A high osbond:DHA ratio is considered a marker of functional DHA deficiency [35–37]. We noted higher concentrations of osbond acyl moieties in liver, plasma, heart and skeletal muscle of SL mice. High osbond:DHA ratios were observed in the aforementioned tissues of SL mice. As the used diet did not contain osbond nor DHA moieties, the mice rely on its endogenous synthesis from dietary linoleic (18:2ω6) and α-linolenic acid (18:3ω3), respectively. As ω6 and ω3 metabolism shares enzymes, we interpret the higher concentration of osbond acyl moieties as an endogenous attempt to synthesize (among others) DHA.

Considering that DHA can dampen fibrotic and inflammatory responses [22], we speculate that its endogenous synthesis is higher in SL mice, possibly for the purpose of generating anti-inflammatory ligands, such as resolvins [38]. Therefore, the data seem to suggest that the synthesis of long-chain polyunsaturated fatty acids (LC-PUFAs) is activated secondary to the inflammatory process.

The plasma BA levels seen in SL mice were (much) higher than what can be expected in non-diseased pre- and postprandial human plasma [39]. Minor experimental differences, such as variations in fasting times (0, 4 or 9 hours fasting) cannot be held responsible for the profoundly elevated plasma BA levels [39]. Plasma lipids, in particular the triglyceride levels, may be mildly elevated in the postprandial state [40]. When comparing unfasted PN21 to (for instance) 4 h fasted PN42, plasma triglyceride levels tend to show higher levels and a higher variability in the unfasted mice (S5 Fig). Other parameters, such as wet liver weight or the relative composition of essential fatty acids (dependent on slow multi-step enzymatic elongations and desaturations [41]) are not expected to differ much between unfasted and 9 h fasted mice.

Liver fibrosis and bile duct proliferation, observed in SL, are elements observed during the onset of certain types of cholestasis [25, 42]. Cholestasis is the reduction or stagnation of bile secretion and bile flow. The accumulation of bile acids (BA) in the plasma, seen in SL mice, is typical for cholestasis [43]. We did not measure bile flow, thus cannot conclusively state that cholestasis is part of the SL phenotype. Alternatively, elevated plasma BA levels can be a consequence of a deficiency in basolateral BA transporters, such as an NTCP deficiency [44, 45]. We did not assess transhepatic BA fluxes, thus cannot conclusively say whether these fluxes are lower in SL mice. Expression levels of *Bsep* and *Ntcp* were lower in the livers of SL mice, but these transporters are known to have an excess transport capacity [45]. Plasma BA composition in SL mice indicated a high abundance of primary BA species. Primary BA species, such as cholate and chenodeoxycholate, are synthesized by the liver and excreted into the bile. In the gut, primary BA are partially converted by microbiota, into secondary BA species, such as deoxycholate and lithocholate. The main enteral BA uptake transporter, ASBT, does not have a strong substrate specificity for conjugated primary or secondary BA [46]. Thus, the lower relative secondary BA abundance may be indicative of either a lower microbial biotransformation capacity [47, 48], or less exposure of the BA pool to the microbiota. The latter could indicate that less BA are excreted into the bile, or that the time BA spend inside the lumen (exposed to the microbiota) is shorter in SL mice. Under physiological conditions, the terminal ileum reabsorbs BA and thereby releases FGF15 protein into the portal circulation which exerts a negative feedback signal to the liver for BA synthesis [49, 50]. BA and FGF15 downregulate *de novo* BA synthesis *via Fxr* and *Shp* and *via* FGFR4 [49, 50], respectively. The lower *Shp* and higher *Cyp7a1* expression levels suggest that FXR signaling is lower in SL mice [50]. The (much) higher relative and absolute tauro-beta-muricholic acid (Tβ-MCA) plasma levels in SL mice (Fig 4) may inhibit FXR signaling [50]. Tβ-MCA is a powerful (gut microbiota-sensitive) FXR antagonist in mice [50]. Considering the high plasma BA levels, the low abundance of secondary BA species, the high hepatic expression of BA synthesis genes, and the low expression of hepatic BA transporters, it is tempting to speculate that the transhepatic (and enterohepatic) BA flux is lower in SL mice.

Liver size is, at least in part, regulated by FGF15-Hippo signaling along the gut-liver axis [23]. FGF15 suppresses YAP signaling by activating Hippo signaling [23]. When YAP signaling is not suppressed, it upregulates genes necessary for bile duct and hepatocyte proliferation [23]. Our data suggest YAP signaling is activated in the livers of SL mice (Fig 4E), suggesting that the low liver weight in SL mice is likely not caused by a suppression of YAP signaling. Instead, activated YAP signaling may be in accordance with the higher rates of hepatocyte cell division seen in the livers of SL mice.

High levels of acylcarnitines have been described in liver biopsies from NASH, but not from NAFLD patients, which has been linked to mitochondrial dysfunction [51]. We noted substantially higher acylcarnitine species in the livers of SL mice, as well as higher hepatic expression of *Pgc1a* (the master regulator of peroxisomal and mitochondrial biogenesis) and *Cpt1a* (part of the carnitine shuttle, rate-limiting factor in β-oxidation). It is tempting to speculate that the inflammatory process exerts a relatively high energy demand, for example generated via mitochondrial β-oxidation. The apparent accumulation of acylcarnitine species, however, may indicate that β-oxidation or the tricarboxylic acid (TCA) cycle is not operating optimally in the livers of SL mice [51]. It could be that β-oxidation outpaces the TCA cycle, upon which incompletely oxidized acyl-carnitine intermediates can accumulate [52]. Alternatively, higher levels of acetylcarnitine may reflect higher peroxisomal β-oxidation rate [53]. Upon impairment of the carnitine shuttle, fatty acids are directed to the peroxisomes for oxidation [53]. The higher hepatic acylcarnitines levels coincided with higher expression markers of the mitochondrial unfolded protein response (*Atf5* and *Ddit3*, higher expressed in the livers of SL mice) [54], which suggest mitochondrial dysfunction [22, 55]. A potential contributor to mitochondrial dysfunction may be the absence of Nicotinamide Nucleotide Transhydrogenase (*Nnt*), which generates the antioxidant compounds glutathione and thioredoxin in the mitochondria [19]. *Nnt* is defective in C57BL/6J [19], but it appeared intact in NL and SL C57BL/6JOlaHsd mice (S4 Fig), and is therefore not likely to contribute to the observed SL phenotype.

We describe a spontaneous, but likely pathological liver phenotype in C57BL/6JOlaHsd mice, bred and reared on a semisynthetic control diet (AIN-93G). Rodent diets are typically available in 'growth' (AIN-93G) and 'maintenance' (AIN-93M) formulation [4]. The "G" variant was formulated to be suitable for growth, pregnancy and lactation [4]. The G and M formulations differ in composition, namely higher protein (20% versus 14%), fat (7% versus 4%) and minerals (mainly calcium carbonate and ferric citrate) in the former, at the expense of corn starch and maltodextrin [4]. Of note, the AIN-93 macro and micro-nutrient composition formulations [4] serve as reference formulations for other semisynthetic diets including ubiquitously used (semisynthetic) high-fat diets and their low-fat controls. We cannot conclude that the choice of the diet is an important factor for the development of the SL phenotype. We observed the SL and NL phenotype in independent cohorts fed AIN-93G from 2 different vendors. Experimental and nutritional conditions were identical for the mice within each cohort. In our view, this makes an environmental or nutritional origin of the SL phenotype unlikely. Though, we cannot exclude that our experimental conditions amplified an otherwise unremarkable genetic heterogeneity. Mice fed AIN-93G compared to a non-purified (chow) control diet, for unknown reasons, have a lower liver weight [56]. Breeders were obtained from a commercial specific-pathogen-free stock, and tested negative for major bacterial, viral and parasitic etiologic agents. We consider it unlikely that the SL phenotype is caused by an infection; we did observe the two phenotypes also within individual cages, housing different mice. The SL and NL phenotype appeared randomly distributed across (surrogate) dams. Due to randomization at PN2, we do not know whether SL/NL is linked to intrauterine conditions. Due to observation of the two phenotypes both in pair-housed and in solo-housed mice, differences in housing conditions or in social hierarchy do not seem to be a contributing factor. Through phenotyping, we were able to identify (an early form of) the SL phenotype at PN42. Such a distinction could not yet be made at weaning (PN21). However, the absence of a phenotype (at PN21) does not fully rule out that the cause for the phenotype is not present. The immediate post-weaning difference in food efficiency ratio and lean mass growth suggests that the cause of the SL phenotype is actually present at weaning. We speculate that the SL phenotype is caused by (subtle) early-life events or variations in C57BL/6JOlaHsd's (epi)genome. As the

occurrence rates varied between cohorts; *i.e.* PN42 (~29%), PN63 (~53%) and PN70 (~36%), it is tempting to speculate that (an) uncontrolled variable(s) potentiate(s) the cause.

The biochemical differences between SL and NL mice resemble, to some extent, (the development of) liver cirrhosis or an early stage of liver failure in humans. Our histological analyses in mice, however, did not (yet) provide clear evidence of liver cirrhosis at PN63 (Fig 2). Future extended longitudinal studies in mice could demonstrate whether or not the SL phenotype evolves to liver cirrhosis and ultimately to end-stage liver failure and early mortality. As liver function appeared impaired, at least in regard to VLDL secretion (Fig 1), we speculate that SL mice may cope less effectively with (models of human) diseases and with the conditions that are associated with ageing. It was beyond the scope of the present study to completely assess whether the same phenomenon also occurs in other mouse (sub-) strains and/or under other nutritional conditions. We observed the SL phenotype in C57BL/6JOlaHsd mice fed semisynthetic AIN-93G. A similar phenotype appears to occur in C57BL/6J mice fed semisynthetic low-fat control diet (S6 Fig). A similar phenomenon appears to have been described in apoE*3-L.CETP mice (C57BL/6J background strain) fed chow [24]. In a large phenotyping study comprising 44 inbred mouse strains (Paigen1, The Jackson Laboratory), challenged to a high fat (semisynthetic) lithogenic diet for 8 weeks, plasma bile acids showed large variability between strains: 0.9 to 350 µM but also within strains [57]. In Paigen1, male C57BL/6J liver weight distribution was bimodal ($p < 0.05$), whereas other strains did not show this either due to low statistical power or absence of bimodality *per se* [57]. The cohorts PN63 and PN70 were highly similar in the assessed parameters (Fig 6F, S2 Fig), although these cohorts originated from separate breeders, and had been studied during different years (2017 and 2018, respectively). This suggests that, by keeping experimental conditions similar, the occurrence of the SL phenotype and its metabolic consequences have been highly reproducible and that it is unlikely that incidental environmental (stress) factors are a major contributing factor to our conclusion. These data suggest that the SL phenotype is likely not specific to our commercial animal supplier, to the 'JOlaHsd' C57BL/6 sub-strain, or to incidental environmental (stress) factors at our animal facility.

Theoretically, the cause of the SL phenotype could still be due to specific environmental conditions present at our facility, which could then affect each new imported cohort. Single-center experiments, like the ones described here, are more vulnerable to biases and methodological pitfalls compared to multi-center experiments [58]. The ApoE*3L.CETP mice results obtained in another facility [24] suggest, however, but do not prove, that the current observation are not unique for our facility. In addition, why such an environmental factor would only affect *some* mice of a cohort, but not others, would then still need to be resolved. It is therefore not (yet) possible to generalize our observations and characterisations to all C57BL/6JOlaHsd mice or indeed to the ubiquitously used C57BL/6J sub-strain.

We identified markers for the identification of the SL phenotype that could potentially be used at postnatal day 42 and later ages. Plasma triglycerides, total cholesterol, bile acid levels or composition, hepatic FAME osbond:DHA ratio, C5DC or C6DC acylcarnitine species, or *Col1a1* gene expression (or other fibrogenesis markers) could serve as potential markers for the (early identification) of the SL phenotype. We were not able to pinpoint any one cause for the SL phenotype and this would certainly have further strengthened our study. Our data (Fig 6) suggest that the cause of the SL phenotype is present at weaning (PN21). We were not able to rule out a genetic origin. If the SL phenotype is, in fact, caused by a genetic variation or defect, then the identification of such a locus by means of genotyping would aid in the early (even pre-experiment) exclusion of those mice from experiments. Societal discussions surrounding the usage of animals in medical research have emphasized the importance of the '3Rs': replacement, reduction, and refinement. The presence of (congenital) defects, which

cause high levels of phenotypic heterogeneity, in otherwise highly homogenous mouse strains is potentially detrimental to the quality of preclinical studies [30, 59]. Reducing phenotypic heterogeneity within cohorts of C57BL/6 mice, or at least the identification of distinct sub-groups within cohorts [24, 59], would contribute to the overall quality of animal research and to the more efficient usage of mice, time and resources.

In summary, our data show that approximately one-third of C57BL/6JOlaHsd mice fed semisynthetic diet develop spontaneous liver disease. This correlates with low liver weight, low VLDL secretion, high plasma bile acids, liver steatosis, fibrosis and inflammation, and mito-chondrial dysfunction. We advise researchers who use this strain (and possibly other C57BL/6 sub-strains, including C57BL/6J) to be aware of this spontaneous phenotype that may interfere with the interpretation of results. Our data suggest that lower plasma triglyceride and choles-terol concentrations (among others) may function as surrogate parameters for early identifica-tion of mice with the apparent SL phenotype.

## Supporting information

**S1 Fig. Plasma, cardiac and skeletal muscle fatty acyl-chain profiles in SL and NL mice.** The plasma FA profile is shown as the fold-change of the percentages *versus* NL (A). The molar plasma triene/tetraene ratio is calculated from $\Sigma(18:3\omega3 + 18:3\omega6 + 20:3\omega6 + 20:3\omega9) / \Sigma(20:4\omega6 + 22:4\omega6)$. Plasma DHA status was calculated as the molar ratio $22:5\omega6/22:6\omega3$ and shown for PN70 (B) and PN63 (J). The cardiac (C), gastrocnemius muscle (F) and hepatic (H) FA profile is shown as the fold-change of absolute values *versus* NL. The DHA status is calcu-lated as the molar OsbA:DHA ratio for cardiac (D), gastrocnemius muscle (E) and hepatic tis-sue (H). Principal component analysis (PCA) on hepatic fatty acyl-chain profiles from pair-housed PN21 (n = 12), PN42 (n = 14), PN63 (n = 14) and PN70 (n = 14) cohorts. Principal component (PC) 1 (42%) and PC 2 (24%) on the y- and x-axis respectively (G). A-B: NL: n = 9, SL: n = 5, data represent the pair-housed PN70 cohort. C-F, H- = I: NL: n = 7–8, SL: n = 6–8, data represent the pair-housed PN63 cohort. Exact two-sided Mann-Whitney U test *: p<0.05, ** p<0.01, *** p<0.001. n.d.: not detected.
(TIF)

**S2 Fig. Classical hierarchical cluster analysis on relative liver weight and assessed plasma and hepatic lipid parameters of the pair-housed PN21, PN42, PN63 and PN70 cohorts.** Heatmap containing the normalized relative liver weight, plasma parameters (lipids, total pro-tein, liver enzymes, bile acids) and hepatic fatty acyl-chain profiles. Each column contains data from one mouse. Each row represents a discrete biometric or biochemical parameter. Each square in the heatmap represents a normalized value (parameter value divided by the average of that parameter in all animals). Missing values are shown as an X in grey. Red color indicates the values is higher than the average, whereas blue color indicates the value is lower than the average of all mice for that particular parameter. Hierarchical clusters were computed using the unweighted pair group method with arithmetic mean (UPGMA) on the Gower's similarity coefficient for mixed data. Data represent the pair-housed PN21 (n = 12), PN42 (n = 14), PN63 (n = 15) and PN70 (n = 14) cohorts. Cophenetic correlation coefficient = 0.842.
(TIF)

**S3 Fig. Hepatic mRNA expression of genes involved in fibrogenesis and lipid metabolism at PN42 and hepatic fatty-acyl chain profiles at PN42 and PN21.** Mice dissected at postnatal day (PN) 42 were split into clusters 1 and 2 based on principal component analysis (Fig 6F). Cluster 1 was positioned near the NL clusters of PN63 and PN70 mice, whereas cluster 2 was positioned near the SL clusters of PN63 and PN70. Hepatic gene expression for fibrogenesis

markers (A) and genes involved in lipid metabolism (B). Hepatic gene expression was corrected for *36b4* and shown as fold-change *versus* Cluster 1. The hepatic FA profile at PN42 is shown as the fold-change of absolute values *versus* Cluster 1 (C). Hepatic DHA status at PN42 (D) and PN21 (F) was calculated as the molar ratio 22:5ω6/22:6ω3. The hepatic FA profile at PN21 is shown as normalized values (each value divided by the average of all values for that particular parameter). Data are shown as Tukey box plots and scatter plots. Data represent the pair-housed PN21 (n = 12) and PN42 (n = 14) cohorts. A-D: cluster 1: n = 10, cluster 2: n = 4. Exact two-sided Mann-Whitney U test *: p<0.05, ** p<0.01.
(TIF)

**S4 Fig. *Nnt^C57BL/6J* genotype in C57BL/6JOlaHsd and C57BL/6J controls.** Pair and solo-housed mice dissected at postnatal day (PN) 70 and random C57BL/6J control samples were genotyped for the spontaneous intragenic deletion mutation in the nicotinamide nucleotide transhydrogenase (*Nnt*) gene. Samples with a band at 743 bp were considered mutant (*Nnt^C57BL/6J*), whereas samples with a band at 579 bp were considered wild-type (*Nnt wild-type*). Lane 1 and 28 contain 3 μl DNA ladder (NEB N0556S). Data represent the pair-housed and solo-housed PN70 cohort. Numbers along the ladder indicate fragment length in base pairs.
(JPG)

**S5 Fig. Plasma lipids at weaning (PN21), PN42 and PN63.** Plasma triglycerides (TG), total (TC), free cholesterol (FC), esterified cholesterol (CE), and non-esterified fatty acids (NEFA) at postnatal day (PN)21 (A), PN42 (B) and PN63 (C). Levels are expressed as absolute concentrations. Mice dissected at PN42 were split into clusters 1 (NL) and 2 (SL) based on principal component analysis (Fig 6F). Data represent the pair-housed PN21 (n = 12 total), PN42 (n = 10 cluster 1, n = 4 cluster 2), and PN63 (n = 7 NL, n = 7 SL) cohorts. Exact two-sided Mann-Whitney U test *: p<0.05, ** p<0.01, *** p<0.001.
(TIF)

**S6 Fig. Classical hierarchical cluster analysis on relative liver weight and assessed plasma lipid parameters of a pair-housed WT C57BL/6J cohort.** WT C57BL/6J mice, reared on chow, were fed a low-fat semisynthetic control diet (D12450J) from PN56 until PN140. The mice were sacrificed at PN140. Heatmap containing the normalized relative liver weight and plasma lipids and total bile acids. Each column contains data from one mouse. Each row represents a discrete biometric or biochemical parameter. Each square in the heatmap represents a normalized value (parameter value divided by the average of that parameter in all animals). Red color indicates the values is higher than the average, whereas blue color indicates the value is lower than the average of all mice for that particular parameter. Hierarchical clusters were computed using the unweighted pair group method with arithmetic mean (UPGMA) on the Gower's similarity coefficient for mixed data (A). Data represents the pair-housed WT C57BL/6J cohort. Cophenetic correlation coefficient = 0.897. Histological staining using hematoxylin and eosin ('H&E') of sample # 8 (B), #11 (C), #12 (D) and #34 (E). Numbers in A correspond to the numbers in B-E. Histology scale bars: 50 μm.
(TIF)

**S7 Fig. Study design from Postnatal Day (PN) 0 to 70.** C57BL/6JOlaHsd mice were bred in-house and nests were culled to 4 males and 2 female pups at PN2. Male pups were weaned at PN21 and either pair-housed (entire PN42 cohort, entire PN63 cohort, pair-housed PN70 cohort) or solo-housed (solo-housed PN70 cohort only). Mice were sacrificed at PN21, PN42, PN63 or PN70. In a subset of the PN70 cohort, upon intraperitoneal injection of the lipoprotein lipase inhibitor poloxamer-407, retro-orbital blood was drawn at 0, 1, 2, 3 & 5 h for

determining the Very-low density lipoprotein-triglyceride (VLDL-TG) secretion rate. PN21: n = 12, PN42: n = 14, PN63: n = 15, PN70, pair-housed: n = 14, PN70, solo-housed: n = 8, PN70, pair-housed, VLDL-TG experiment: n = 14, PN70, pair-housed, VLDL-TG experiment: n = 8.
(TIF)

**S8 Fig. Plasma samples at PN70, made from cardiac puncture blood samples.** Each tube contained exactly 400 μl of plasma and was not diluted. The left sample (above the '5') was obtained from a mouse which was later labeled "SL", whereas the right sample (above the '6') was obtained from a mouse which was later labeled "NL".
(TIF)

**S1 Raw image.**
(TIF)

**S1 Table. Liver acylcarnitine species in SL and NL mice at PN42.** Mice dissected at PN42 were split into clusters 1 (NL) and 2 (SL) based on principal component analysis (Fig 6F). Liver (nmol/g) acylcarnitine species are shown as absolute values. Values represent means and SD. Data represent the pair-housed PN42 cohort. Exact two-sided Mann Whitney U test. n.s.: not significantly different between NL and SL.
(DOC)

**S2 Table. Primer and TaqMan probe sequences.**
(DOC)

**S3 Table. Fatty acid composition of the AIN-93G diet.** *: all in FA weight%.
(DOC)

**S4 Table. Calculated nutrient composition (in g/kg) of the diets.**
(DOC)

**S5 Table. Liver acylcarnitine species at PN21.** Mice dissected at PN21 did not show clear clusters, based on principal component analysis (Fig 6F). Liver (nmol/g) acylcarnitine species are shown as absolute values for each mouse.
(DOC)

## Acknowledgments

The authors would like to thank Renze Boverhof, Michelle Brad and Dilys Eikelboom (Department of Pediatrics, University Medical Center Groningen) for technical assistance. We thank prof. Gertjan van Dijk (Behavioral Neuroscience, Groningen Institute for Evolutionary Life Sciences, University of Groningen) for valuable comments. We thank prof. Bert K. Groen (Amsterdam Diabetes Center and Department of Vascular Medicine, Academic Medical Center, Amsterdam, the Netherlands) for valuable insights. We thank Johanneke van der Harst, Lidewij Schipper and Maryam Rakhshandehroo (Danone Nutricia Research, the Netherlands) for helpful comments on the manuscript.

## Author Contributions

**Conceptualization:** Onne A. H. O. Ronda, Mirjam H. Koster, Folkert Kuipers, Henkjan J. Verkade.

**Formal analysis:** Onne A. H. O. Ronda, Alain de Bruin, Rachel E. Thomas.

**Funding acquisition:** Bert J. M. van de Heijning, Henkjan J. Verkade.

**Investigation:** Onne A. H. O. Ronda, Alain de Bruin, Rachel E. Thomas, Ingrid Martini, Martijn Koehorst, Albert Gerding, Mirjam H. Koster, Vincent W. Bloks, Angelika Jurdzinski, Niels L. Mulder, Rick Havinga.

**Visualization:** Onne A. H. O. Ronda, Alain de Bruin, Rachel E. Thomas.

**Writing – original draft:** Onne A. H. O. Ronda.

**Writing – review & editing:** Bert J. M. van de Heijning, Eline M. van der Beek, Dirk-Jan Reijngoud, Folkert Kuipers, Henkjan J. Verkade.

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
