## [Decision Letter · Decision Letter 0]

1 Jun 2020

PONE-D-20-09497

Spontaneous liver disease in wild-type C57BL/6JOlaHsd mice fed semisynthetic diet

PLOS ONE

Dear Dr. Ronda,

Thank you for submitting your manuscript to PLOS ONE. After careful consideration, we feel that it has merit but does not fully meet PLOS ONE’s publication criteria as it currently stands. Therefore, we invite you to submit a revised version of the manuscript that addresses the points raised during the review process.

Thank you for submitting your manuscript to PLOS ONE. After careful consideration, we feel that it has merit but does not fully meet PLOS ONE’s publication criteria as it currently stands. Therefore, we invite you to submit a revised version of the manuscript that addresses the points raised during the review process.

Our ability to obtain multiple reviews for your manuscript has been hampered by the COVID-19 pandemic. However, we have been able to provide you will a thoughtful review that raises a number of minor and major concerns that need to be addressed. In the sake of timeliness, we are allowing you to revise your manuscript based the current review. 

Please review your cover letter prior to submission. Your original submission contained a cover letter that appeared to be addressed to the journal Laboratory Investigation.

We look forward to receiving your revised manuscript.

Kind regards,

Michael W Greene, Ph.D.

Academic Editor

PLOS ONE

Journal Requirements:

2. We note that you reference both an animal experiment committee and a local animal welfare body. Please clarify which is your Institutional Animal Care and Use Committee and ensure that the full name of this committee is in your ethics statement and Methods section.

3. To comply with PLOS ONE submissions requirements, in your Methods section, please provide additional information on the animal research and ensure you have included details on the specific efforts made to alleviate suffering."

4. Please report the actual p-values in Table 2 instead of "n.s.".

5. Thank you for submitting your manuscript to PLOS ONE. Our journal requires that methods are described in enough detail to allow suitably skilled investigators to fully replicate your study. Please provide more information on the sources of all reagents, proteins, antibodies, and equipment in the methods section of your manuscript. If materials, methods, and protocols are well established, authors may cite articles where those protocols are described in detail, but the submission should include sufficient information to be understood independent of these references. We note that some of your methods are reported "as described" without further description in the actual manuscript. Please revise your manuscript so that all protocols are sufficiently described. For more information please see https://journals.plos.org/plosone/s/submission-guidelines#loc-materials-and-methods.

7. Thank you for stating the following in the Competing Interests section:

'The authors have declared that no competing interests exist.'

We note that one or more of the authors are employed by a commercial company: Danone Nutricia Research.

Additional Editor Comments (if provided):

Reviewers' comments:

Reviewer's Responses to Questions

**Comments to the Author**

1. Is the manuscript technically sound, and do the data support the conclusions?

Reviewer #1: Partly

2. Has the statistical analysis been performed appropriately and rigorously? 

Reviewer #1: Yes

3. Have the authors made all data underlying the findings in their manuscript fully available?

Reviewer #1: Yes

4. Is the manuscript presented in an intelligible fashion and written in standard English?

Reviewer #1: Yes

5. Review Comments to the Author

Reviewer #1: The manuscript is well written and describes extensive, and well done phenotypic investigation of a unique variant in a common mouse model fed a semi-synthetic diet.

The authors have done extensive work evaluating a spontaneous phenotype associated with feeding a semisynthetic diet in their C57BL/6JOlaHsd mouse colony.  They do an excellent job describing the changes in liver weight and body weight as well as some biochemical and metabolic features of their small liver phenotype and they demonstrate that these changes are independent of dominance behavior.  I commend the thorough histologic pathology evaluation.  The authors draw fairly broad conclusions from their findings, and they suggest that this phenotype extends to all C57BL/6JOlaHsd mice fed this specific diet. There is discussion on the impact of other unique metabolic phenotypes observed in other stains and the authors suggest that plasma lipids may be used a biomarker for the small liver phenotype.  While this paper is overall well written and has extensive analysis, I have a few concerns.

The rationale and effects of the fasting time are a bit confusing, and if they are not consistent, would have major metabolic effects.  It looks like the PN21 group was not fasted since they were sacrificed at weaning, but this means they were collected following a lipid rich diet.  What diet were the Dams fed? It appears the PN42 were fasted for 4 hours during a time where mice would not be expected to eat much (9 am to 1 pm), while the remaining two groups (PN63 and PN70) were fasted for 8 hours during a prime feeding time for the mice (12 am – 9 am).  The overall objective of the study is not directly comparing between the time points, but this may affect some of the outcomes in the section that reviews the pathogenesis of the process.  IN some areas of the manuscript the authors make direct comparisons between the cohorts where these differences in fasting are more problematic.  The choice for the fasting protocol should be clarified or justified, or both.In the discussion, the authors consider portosystemic shunt as a possible cause for the SL phenotype, but I am unclear how that relates to diet.  PSS can be intrahepatic or extrahepatic, and I believe are congenital, or acquired secondary to fibrosis or other primary disease processes.  Large extra hepatic vessels could be identified grossly, but would be congenital and not diet dependent.  Intrahepatic shunts should be able to be identified histologically.  The point of this section in the discussion did not seem relevant and distracted from the findings.Given the presence of a small liver, fibrosis, impaired BA processing and alter lipids, and altered hepatic enzymology there should be a more direct statement regarding overt liver failure and early cirrhosis as opposed to simply calling it a small liver phenotype.I have one major concern: It is unclear to me if this small liver phenotype is present in C57BL/6JOlaHsd mice outside of this particular housing unit.  At the end of the Introduction, the authors describe some of the aspects of the phenotype as a “spontaneous variation” and I think that is an accurate term.  Inbred mouse models have extremely high homozygosity, but they still have variation and are capable of genetic drift and spontaneous polymorphisms that can alter the expected phenotype, particularly within closed breeding groups.  The authors discuss their findings in regards to the significance of this phenotype, but it is not clear in my reading of the manuscript that this phenotype is not limited to this specific colony.  There is discussion as to unique vendors for the food, but I did not see anything in regards to the mice and it appears all cohorts of mice were bred onsite.  If the authors do not have evidence that this phenotype extends beyond their colony, there is very limited impact of all of their hard work in phenotyping the disease/model.  The author suggest this is a developmental feature.  Is there any information on the lifespan of these mice in this colony? Do they eventually segregate into these two phenotypes and does the diet exacerbate an underlying condition, or is the pathogenesis triggered by this diet alone? This last question is likely beyond the scope of the study, but would make for a more interesting and impactful discussion. This study does such wonderful work with the phenotyping of these mice, but the real importance of the changes stem from the underlying cause, which based on the information provided appears to be developmental and likely genetic.  Instead of using lipids post-study it would be more ideal to genotype the mice for this allele prior to putting them on expensive diets.  Please note, while this is a major concern, it does not negate the findings or the extensive characterization of the phenotype.  It just may limit the value of the work to those who work directly with this specific colony. What would be necessary is extensive clarity in the discussion and to some extent the conclusions to this point.

6. PLOS authors have the option to publish the peer review history of their article (what does this mean?). If published, this will include your full peer review and any attached files.

Reviewer #1: No

---

## [Author Response · Author response to Decision Letter 0]

30 Jul 2020

5. Review Comments to the Author

Reviewer #1: The manuscript is well written and describes extensive, and well done phenotypic investigation of a unique variant in a common mouse model fed a semi-synthetic diet.

The authors have done extensive work evaluating a spontaneous phenotype associated with feeding a semisynthetic diet in their C57BL/6JOlaHsd mouse colony. They do an excellent job describing the changes in liver weight and body weight as well as some biochemical and metabolic features of their small liver phenotype and they demonstrate that these changes are independent of dominance behavior. I commend the thorough histologic pathology evaluation. The authors draw fairly broad conclusions from their findings, and they suggest that this phenotype extends to all C57BL/6JOlaHsd mice fed this specific diet. There is discussion on the impact of other unique metabolic phenotypes observed in other stains and the authors suggest that plasma lipids may be used a biomarker for the small liver phenotype. While this paper is overall well written and has extensive analysis, I have a few concerns.

1. The rationale and effects of the fasting time are a bit confusing, and if they are not consistent, would have major metabolic effects. It looks like the PN21 group was not fasted since they were sacrificed at weaning, but this means they were collected following a lipid rich diet. What diet were the Dams fed? It appears the PN42 were fasted for 4 hours during a time where mice would not be expected to eat much (9 am to 1 pm), while the remaining two groups (PN63 and PN70) were fasted for 8 hours during a prime feeding time for the mice (12 am – 9 am). The overall objective of the study is not directly comparing between the time points, but this may affect some of the outcomes in the section that reviews the pathogenesis of the process. IN some areas of the manuscript the authors make direct comparisons between the cohorts where these differences in fasting are more problematic. The choice for the fasting protocol should be clarified or justified, or both.

---------------- AUTHOR RESPONSE:

We agree that the differences in the fasting times merit additional clarification and have added this to the new version of the manuscript. The dams were fed AIN-93G throughout breeding, pregnancy and lactation. We do agree and have now stated that we cannot exclude that differences in fasting may have affected some of the biochemical parameters, such as plasma triglyceride levels. Indeed, these levels show variability and tend to be elevated at PN21 compared to our other cohorts.

However, other parameters, such as liver weight or the relative composition of essential fatty acids (dependent on slow multi-step enzymatic conversions) in the liver are not expected to differ much between unfasted and 9 h fasted mice. The occurrence of a spontaneous divergence in liver weight under the used experimental conditions is not affected by the duration of fasting immediately prior to dissection.

2. In the discussion, the authors consider portosystemic shunt as a possible cause for the SL phenotype, but I am unclear how that relates to diet. PSS can be intrahepatic or extrahepatic, and I believe are congenital, or acquired secondary to fibrosis or other primary disease processes. Large extra hepatic vessels could be identified grossly, but would be congenital and not diet dependent. Intrahepatic shunts should be able to be identified histologically. The point of this section in the discussion did not seem relevant and distracted from the findings.

---------------- AUTHOR RESPONSE:

We agree with the reviewer that we have no data which prove or disprove the occurrence of congenital portosystemic shunting and its possible relationship to the small liver phenotype. Considering this uncertainty, we agree with the reviewer’s opinion that it distracts from the findings. Therefore, we have omitted this section in the new version of the discussion.

3. Given the presence of a small liver, fibrosis, impaired BA processing and alter lipids, and altered hepatic enzymology there should be a more direct statement regarding overt liver failure and early cirrhosis as opposed to simply calling it a small liver phenotype.

---------------- AUTHOR RESPONSE:

Agreed. The biochemical differences between SL and NL mice resemble, to some extent, human liver cirrhosis and the early stages of liver failure. Our histological analyses did not (yet) find clear evidence of cirrhosis at the indicated time points. Indeed, extended longitudinal studies could indicate that the SL phenotype evolves to liver cirrhosis and ultimately to end-stage liver failure and early mortality. 

To honor this valuable remark of the reviewer, we have included this statement in the new version of the manuscript.

4. I have one major concern: It is unclear to me if this small liver phenotype is present in C57BL/6JOlaHsd mice outside of this particular housing unit. At the end of the Introduction, the authors describe some of the aspects of the phenotype as a “spontaneous variation” and I think that is an accurate term. Inbred mouse models have extremely high homozygosity, but they still have variation and are capable of genetic drift and spontaneous polymorphisms that can alter the expected phenotype, particularly within closed breeding groups. The authors discuss their findings in regards to the significance of this phenotype, but it is not clear in my reading of the manuscript that this phenotype is not limited to this specific colony. There is discussion as to unique vendors for the food, but I did not see anything in regards to the mice and it appears all cohorts of mice were bred onsite. If the authors do not have evidence that this phenotype extends beyond their colony, there is very limited impact of all of their hard work in phenotyping the disease/model. The author suggest this is a developmental feature. Is there any information on the lifespan of these mice in this colony? Do they eventually segregate into these two phenotypes and does the diet exacerbate an underlying condition, or is the pathogenesis triggered by this diet alone? This last question is likely beyond the scope of the study, but would make for a more interesting and impactful discussion. This study does such wonderful work with the phenotyping of these mice, but the real importance of the changes stem from the underlying cause, which based on the information provided appears to be developmental and likely genetic. Instead of using lipids post-study it would be more ideal to genotype the mice for this allele prior to putting them on expensive diets. Please note, while this is a major concern, it does not negate the findings or the extensive characterization of the phenotype. It just may limit the value of the work to those who work directly with this specific colony. What would be necessary is extensive clarity in the discussion and to some extent the conclusions to this point.

---------------- AUTHOR RESPONSE:

We thank the reviewer for (also) this indeed important aspect on the generalizability of the findings with respect to the mice and the environment.

The cohorts described in this manuscript were bred from commercially obtained C57BL/6JOlaHsd breeders. These breeders were specifically purchased for this study and did not originate from our local mouse colony. The breeders were used to breed the mice used throughout this manuscript. Only the F1 offspring were used. Separate batches of breeders were ordered from a commercial supplier in the years 2016, 2017 and 2018. Each batch was used separately to breed pups to be used in this study.

The cohorts PN63 and PN70 were highly similar in the assessed parameters, although these cohorts originated from separate breeders, and had been studied during different years. This suggests that, by keeping experimental conditions similar, the occurrence of the SL phenotype and its metabolic consequences have been highly reproducible.

The mice were obtained from a commercial breeder which had the resources to carefully monitor and control health, reproduction, and genetic drift. The usage of a large number of breeders, obtained in separate batches across separate years gives, in our view, a fair representative of what one may expect from a commercial animal supplier. This approach, however, does not completely exclude the possibility of a genetic cause. It could be possible that the ‘JOlaHsd’ sub-strain experienced genetic drift away from the ‘J’ sub-strain. To investigate whether a similar phenomenon occurs in the ‘J’ sub-strain, we briefly analysed a ‘J’ cohort at our own facility. This cohort was not obtained from the commercial animal supplier used previously. In S6 Fig, we showed that a similar phenomenon does appear to occur in C57BL/6J mice. In the ApoE*3L.CETP model, which is used both in our facility but also elsewhere, a comparable phenotypic heterogeneity has been extensively described, although not characterized as well as we now have reported for the ‘JOlaHsd’ mice. Recently, this phenotypic heterogeneity has been attributed to the presence of two distinct sub-groups in cohorts, rather than to heterogeneity in the classic sense (Tarasco et al., 2018 Am J Physiol Gastrointest Liver Physiol.).

The ‘non-responders’ (compared to the ‘responders’, as the subgroups have been named) in ApoE*3L.CETP mice appear to have very similar characteristics as our SL phenotype. These data strongly suggest that the SL phenotype is possibly not a ‘JOlaHsd’ idiosyncrasy or a breeder-specific phenomenon. We highly value these comments of the reviewer and have summarized this discussion in the new version of the manuscript.

We do not have information regarding the lifespan of these mice. It is unclear to us whether the diet plays a major (if any) role in the occurrence of the SL phenotype. We agree with the reviewer that understanding whether the diet exacerbates an underlying condition or whether the diet triggers the phenotype per se, would make for a more impactful discussion. We were not able to pinpoint any one cause for the SL phenotype, what would have further strengthened our manuscript. We have added these aspects, and the points above, to the new version of the discussion.

Theoretically, the cause of the SL phenotype could still be due to specific environmental conditions present at our facility, which could then affect each new imported cohort. Single-center experiments are more vulnerable to biases and methodological pitfalls compared to multi-center experiments. The ApoE*3L.CETP mice results obtained in another facility suggest, however, but do not prove, that the current observations are not unique for our facility. In addition, why such an environmental factor would only affect some mice, but not others, would then still need to be resolved. It is therefore not (yet) possible to generalize our observations and characterisations to all C57BL/6JOlaHsd mice or indeed to the ubiquitously used C57BL/6J sub-strain. We have also added these points to the new version of the discussion.

---

## [Decision Letter · Decision Letter 1]

18 Aug 2020

Spontaneous liver disease in wild-type C57BL/6JOlaHsd mice fed semisynthetic diet

PONE-D-20-09497R1

Dear Dr. Ronda,

We’re pleased to inform you that your manuscript has been judged scientifically suitable for publication and will be formally accepted for publication once it meets all outstanding technical requirements.

Kind regards,

Michael W Greene, Ph.D.

Academic Editor

PLOS ONE

Additional Editor Comments:

I appreciate your effort and attention to detail in the revision. I also appreciate your willingness to proceed with just one reviewer for your manuscript. The COVID-19 pandemic has resulted in real stress in the review process.

Reviewers' comments:

Reviewer's Responses to Questions

**Comments to the Author**

1. If the authors have adequately addressed your comments raised in a previous round of review and you feel that this manuscript is now acceptable for publication, you may indicate that here to bypass the “Comments to the Author” section, enter your conflict of interest statement in the “Confidential to Editor” section, and submit your "Accept" recommendation.

Reviewer #1: All comments have been addressed

2. Is the manuscript technically sound, and do the data support the conclusions?

Reviewer #1: Yes

3. Has the statistical analysis been performed appropriately and rigorously? 

Reviewer #1: Yes

4. Have the authors made all data underlying the findings in their manuscript fully available?

Reviewer #1: Yes

5. Is the manuscript presented in an intelligible fashion and written in standard English?

Reviewer #1: Yes

6. Review Comments to the Author

Reviewer #1: All of the original comments have been sufficiently addressed by the authors in the Comments and Discussion. Thank you.

7. PLOS authors have the option to publish the peer review history of their article (what does this mean?). If published, this will include your full peer review and any attached files.

Reviewer #1: No

---

## [Editor Report · Acceptance letter]

11 Sep 2020

PONE-D-20-09497R1 

Spontaneous liver disease in wild-type C57BL/6JOlaHsd mice fed semisynthetic diet 

Dear Dr. Ronda:

I'm pleased to inform you that your manuscript has been deemed suitable for publication in PLOS ONE. Congratulations! Your manuscript is now with our production department. 

Kind regards, 

on behalf of

Dr. Michael W Greene 

Academic Editor

PLOS ONE